# Phosphorylation of the DNA damage repair factor 53BP1 by ATM kinase controls neurodevelopmental programs in cortical brain organoids

**Bitna Lim**[1☯¤], **Yurika Matsui**[1☯], **Seunghyun Jung**[1], **Mohamed Nadhir Djekidel**[2], **Wenjie Qi**[2], **Zuo-Fei Yuan**[3], **Xusheng Wang**[3], **Xiaoyang Yang**[1], **Nina Connolly**[1], **Abbas Shirinifard Pilehroud**[1], **Haitao Pan**[4], **Fang Wang**[4], **Shondra M. Pruett-Miller**[5], **Kanisha Kavdia**[3], **Vishwajeeth Pagala**[3], **Yiping Fan**[2], **Junmin Peng**[1,6], **Beisi Xu**[2], **Jamy C. Peng**[1]*

1 Department of Developmental Neurobiology, St Jude Children's Research Hospital, Memphis, Tennessee, United States of America, 2 Center for Applied Bioinformatics, St Jude Children's Research Hospital, Memphis, Tennessee, United States of America, 3 Center for Proteomics and Metabolomics, St. Jude Children's Research Hospital, Memphis, Tennessee, United States of America, 4 Department of Biostatistics, St Jude Children's Research Hospital, Memphis, Tennessee, United States of America, 5 Department of Cell & Molecular Biology, St Jude Children's Research Hospital, Memphis, Tennessee, United States of America, 6 Department of Structural Biology, St Jude Children's Research Hospital, Memphis, Tennessee, United States of America

☯ These authors contributed equally to this work.
¤ Current address: CHA Future Medicine Research Institute, Seongnam, Republic of Korea
* jamy.peng@stjude.org

**Data Availability Statement:** All sequencing data are deposited in NCBI GEO database under accession number GSE231321. Codes for

## Abstract

53BP1 is a well-established DNA damage repair factor that has recently emerged to critically regulate gene expression for tumor suppression and neural development. However, its precise function and regulatory mechanisms remain unclear. Here, we showed that phosphorylation of 53BP1 at serine 25 by ATM is required for neural progenitor cell proliferation and neuronal differentiation in cortical brain organoids. Dynamic phosphorylation of 53BP1-serine 25 controls 53BP1 target genes governing neuronal differentiation and function, cellular response to stress, and apoptosis. Mechanistically, ATM and RNF168 govern 53BP1's binding to gene loci to directly affect gene regulation, especially at genes for neuronal differentiation and maturation. 53BP1 serine 25 phosphorylation effectively impedes its binding to bivalent or H3K27me3-occupied promoters, especially at genes regulating H3K4 methylation, neuronal functions, and cell proliferation. Beyond 53BP1, ATM-dependent phosphorylation displays wide-ranging effects, regulating factors in neuronal differentiation, cytoskeleton, p53 regulation, as well as key signaling pathways such as ATM, BDNF, and WNT during cortical organoid differentiation. Together, our data suggest that the interplay between 53BP1 and ATM orchestrates essential genetic programs for cell morphogenesis, tissue organization, and developmental pathways crucial for human cortical development.

analyzing sequencing data are deposited in https://doi.org/10.6084/m9.figshare.7411835. Mass spectrometry data were deposited in ProteomXchange, with project accession number PXD041699. Numerical data are in S1 Data, and uncropped Western blot images are in S1 Raw Images.

**Funding:** This work was supported by the American Lebanese Syrian Associated Charities (https://www.stjude.org/ to JCP), American Cancer Society (https://www.cancer.org/; 132096-RSG-18-032-01-DDC to JCP), and NIH (https://www.nih.gov/; 1R01GM134358-05 to JCP). The funders had no role in study design, data collection and analysis, decision to publish, or preparation of the manuscript.

**Competing interests:** The authors have declared that no competing interests exist.

**Abbreviations:** ATM, ataxia telangiectasia mutated; BSA, bovine serum albumin; FDR, false discovery rate; GO, Gene Ontology; GSEA, gene set enrichment analysis; hESC, human embryonic stem cell; KO, knockout; NPC, neural progenitor cell; PSM, peptide spectral match; TMT LC-MS/MS, liquid chromatography-tandem mass spectrometry; WB, western blot; WT, wild type; 53BP1, p53 binding protein 1; 53BP1-pS25, 53BP1 phosphorylated at serine 25.

## Introduction

Transcription ensures the proper expression of genetic information for the development and function of the organism, whereas DNA repair maintains the integrity of the genetic code. These 2 processes share cross-functional factors, including CSB, TFII, and XPG, which repair DNA damage caused by torsional stress from transcription-initiating RNA polymerase II [1–3]. Conversely, some proteins initially believed to function exclusively in DNA repair have been found to regulate gene expression. For example, 53BP1 (p53 binding protein 1) is a key regulator of DNA repair mechanisms, promoting nonhomologous end-joining over homologous recombination [4]. During the DNA damage response, 53BP1 plays a pivotal role in p53-mediated activation of tumor suppressive genetic programs [5]. Recent research has also revealed that 53BP1 collaborates with the chromatin modifier UTX in neural progenitor cells (NPCs), promoting an open chromatin to facilitate the activation of neurogenic or corticogenic programs [6]. Intriguingly, the 53BP1–UTX interaction is observed in humans but not in mice [6]; the mechanism is not well conserved and regulates primate neurodevelopment. These discoveries highlight the importance of 53BP1 in gene regulation for tumor suppression and neural development. However, the precise mechanisms underlying 53BP1's role in gene regulation and its upstream mechanism are yet to be fully understood.

Studies of 53BP1 have primarily focused on its role in the DNA damage response. To localize to chromatin with double-stranded breaks, 53BP1 uses its BRCT domain to bind to γH2AX, the Tudor domain to bind to H4K20 dimethylation, and its UDR segment to bind to ubiquitinated H2AK15 [7–10]. Additionally, the phosphorylated SQ/TQ motif of 53BP1 coordinates the docking of RIF1 or SCAI, selectively promoting nonhomologous end-joining or reducing homologous recombination [11,12]. These interactions are likely relevant to the gene regulatory activities of 53BP1. For example, γH2AX recruits 53BP1 and is required for resolving R-loops, DNA demethylation, transcription activation, and transcription elongation [13,14]. These findings suggest that the activities of 53BP1 in DNA damage response are interconnected with its gene regulatory functions.

The studies mentioned above have contributed to a model of 53BP1, where posttranslational modifications of its different residues and domains coordinate various activities. Most prominently, numerous residues of 53BP1 are phosphorylated by ATM (ataxia telangiectasia mutated) kinase [10,15–17]. ATM-mediated phosphorylation of 53BP1 or 53BP1-interacting proteins controls protein interactions, cellular localization, and DNA repair mechanisms [11,12,18–20]. Despite these discoveries, the impact of phosphorylation on the gene regulatory activity of 53BP1 remains unknown. Here, we report that phosphorylation of 53BP1-serine 25 by ATM is crucial for the proper expression of genetic programs during the growth and development of cortical brain organoids. ATM-dependent phosphorylation controls the chromatin binding of 53BP1 to genomic targets functioning in several key pathways, including neuronal differentiation, cytoskeleton, p53, and ATM, BNDF, and WNT signaling pathways. These results highlight the essential role of 53BP1 phosphorylation in regulating genetic programs for the differentiation of cortical brain organoids.

## Results

### Phosphorylated 53BP1-S25 increases during differentiation of hESCs into NPCs

Although 53BP1 is required for human embryonic stem cells (hESCs) to differentiate into NPCs [6], its levels did not change during differentiation (Fig 1A). Therefore, its regulation is likely posttranslational during neural differentiation. Human NPCs were analyzed by RNA-

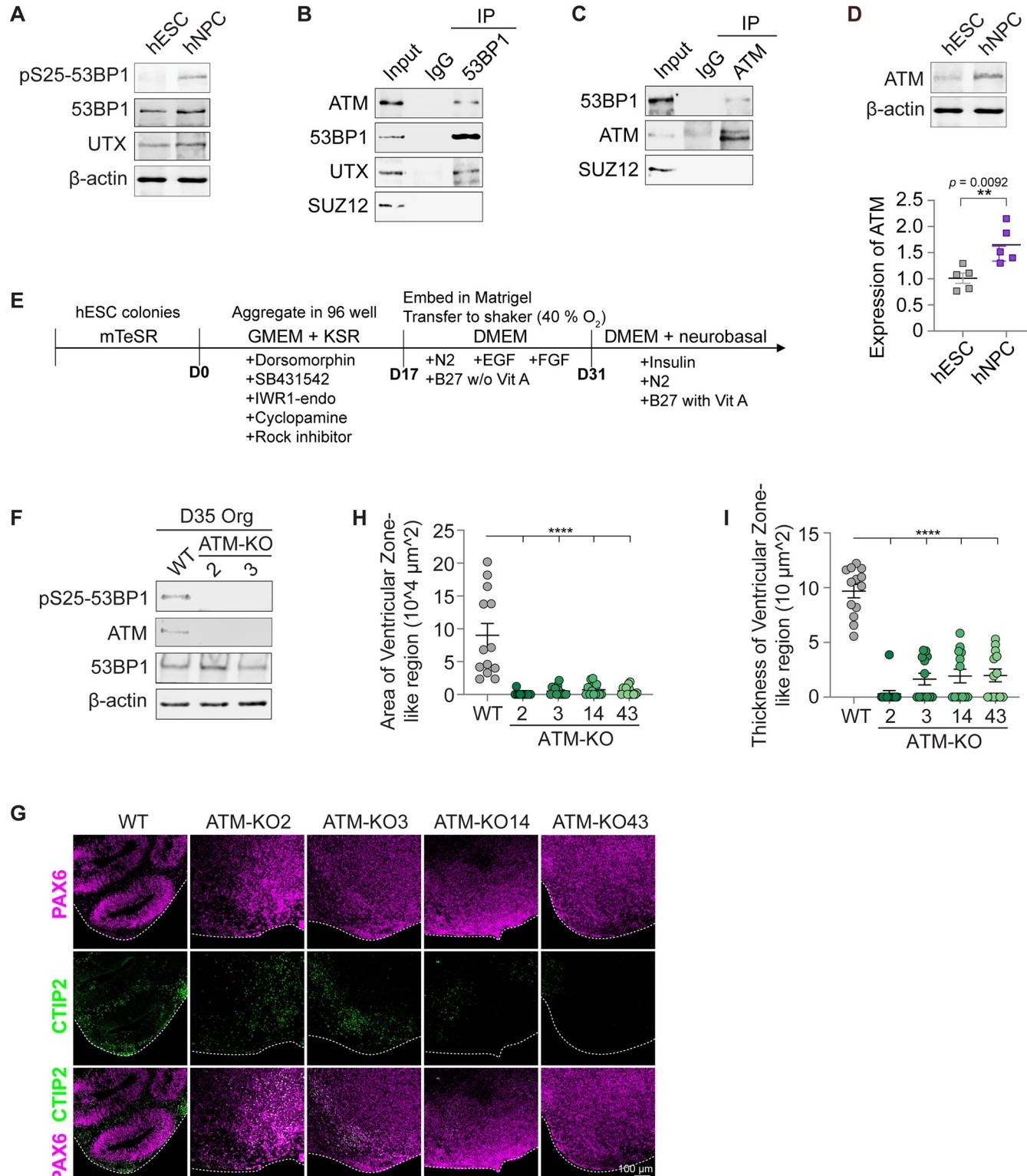

**Fig 1. ATM binds 53BP1, is required for pS25-53BP1, and promotes cortical organoid differentiation.** (**A**) WB of the nuclear extract of hESCs and hNPCs showed marked increase of 53BP1-pS25 in hNPCs. WB analysis of IgG, (**B**) 53BP1, and (**C**) ATM co-immunoprecipitation in the nuclear extract of hESCs. (**D**) Quantification of the relative ATM protein levels (normalized to β-ACTIN) in 5 replicate WB analyses of hESCs and hNPCs. (**E**) Schematic diagram of the cortical organoid differentiation. Aggregates were formed in the induction media for 17 days, embedded in Matrigel droplets and cultured in cortical differentiation medium for 16 days, and then cultured in cortical maturation media thereafter. (**F**) WB analysis of WT and ATM-KO cortical organoids at day

35 of differentiation. Immunofluorescence of (**G**) PAX6 and CTIP2 and (**J**) KI67 in cryosections of cortical organoids at day 35 of differentiation. Bar, 100 μm. At day 35 of differentiation, the (**H**) area and (**I**) thickness of VZ-like regions were compared between groups. Data points represent single organoids. The mean ± SEM values were compared by one-way ANOVA with Dunnett's multiple comparisons test to yield **** indicating $p < 0.0001$. $n = 13$ organoids/group. Underlying numerical values for figures are found in S1 Data. ATM, ataxia telangiectasia mutated; DMEM, Dulbecco's Modified Eagle Medium; GMEM, Glasgow Modified Essential Medium; hESC, human embryonic stem cell; hNPC, human neural progenitor cell; IgG, immunoglobulin G; KO, knockout; KSR, Knockout Serum Replacement; VZ, ventricular zone; WB, western blot; WT, wild type; 53BP1, p53 binding protein 1; 53BP1-pS25, 53BP1 phosphorylated at serine 25.

seq and immunofluorescence to validate successful NPC generation (S1A–S1E Fig). 53BP1 is targeted by various kinases, including ATM, and we hypothesized that 53BP1 phosphorylation regulates the differentiation of hESCs into NPCs. Intriguingly, we found that the levels of 53BP1 phosphorylated at serine 25 (53BP1-pS25) were markedly increased in NPCs compared to hESCs (Figs 1A and S1F–S1H). The levels of the DNA damage marker γH2AX were similar in NPCs and hESCs (S1I Fig), suggesting that the increase in 53BP1-pS25 levels during NPC differentiation is not due to increased DNA damage.

## ATM is required for 53BP1-S25 phosphorylation, cell differentiation, and tissue morphogenesis in cortical organoids

The ATM kinase phosphorylates 53BP1-S25 [15], and we thus investigated whether ATM plays a role in neural differentiation. First, we found that ATM co-immunoprecipitated with 53BP1, as did the positive control UTX, but not with the negative control SUZ12 (a core subunit of PRC2, which does not bind these proteins; Fig 1B). Similarly, 53BP1 co-immunoprecipitated with ATM, but not with the negative control SUZ12 (Fig 1C). Like 53BP1-pS25, ATM levels were significantly increased in NPCs compared with hESCs (Fig 1D). ATM upregulation in NPCs was shown by a previous DNA damage response study [21].

Next, we used the CRISPR-Cas9 system to generate 4 *ATM*-knockout (KO) hESC lines (Figs 2A, 2B, and S1J). Data from RNA-seq and immunofluorescence showed that *ATM*-KO did not markedly alter hESC pluripotency (S2C and S2D Fig). All cell lines underwent karyotyping analysis and were characterized as karyotypically normal (S1 Table). *ATM*-KO lines had minor abnormalities, as expected due to the requirement of ATM for DNA damage repair. To analyze the role of ATM in human cortical development, we used an established protocol to differentiate wild type (WT) and *ATM*-KO hESCs into cortical organoids (Fig 1E, Methods; [6]). We did not detect 53BP1-pS25 in *ATM*-KO D35 cortical organoids (Fig 1F) nor NPCs (S2B Fig), consistent with loss of ATM-mediated phosphorylation of 53BP1-S25 during neural differentiation of hESCs. *ATM*-KO modestly reduced γH2AX levels in NPCs (S2E Fig), suggesting that ATM promotes the phosphorylation of H2AX-S139 in NPCs.

By day 35 (D35) of differentiation, WT cortical organoids expressed the forebrain NPC marker PAX6 in ventricular zone–like regions that were radially organized (Fig 1G). In contrast, *ATM*-KO D35 cortical organoids displayed disorganized and smaller ventricular zone–like regions (Fig 1G–1I). We quantitatively compared NPC proliferation, neuronal differentiation, cell death, and cell organization in *ATM*-KO versus WT cortical organoids. Examination of endogenous DNA damage, by γH2AX immunofluorescence, did not reveal marked difference (S3A Fig), confirming our western blot (WB) results in S1I Fig. Although quantification of cell death marker cleaved-caspase 3 by FACS revealed a modest increase of cell death in D21 ATM-KO cortical organoids, FACS and immunofluorescence quantification showed that D28 and D35 *ATM*-KO and WT cortical organoids are similar in cell death frequencies (S3B–S3E Fig). Cell proliferation frequencies did not significantly differ between D28 and D35 *ATM*-KO and WT cortical organoids (S4 Fig). We next quantified immature neuronal marker NEUN and PAX6/CTIP2 ratios. Despite lower levels of immature neuronal differentiation, ATM-KO

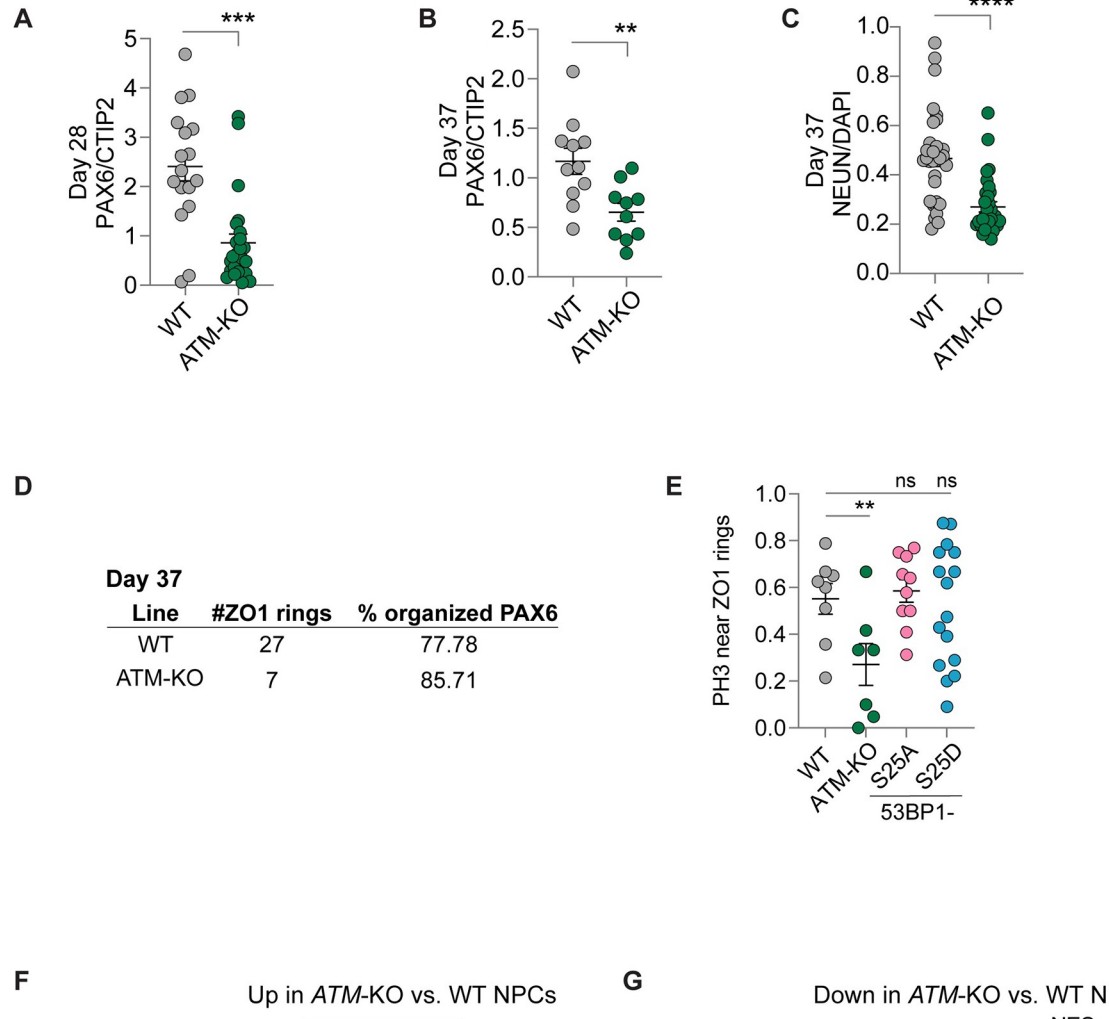

**Fig 2. Analysis of neuronal differentiation and cell organization in cortical organoids.** Quantification of PAX6/CTIP2 ratios in (**A**) D28 and (**B**) D37 cortical organoids. (**C**) Quantification of NEUN/DAPI in D37 cortical organoids. (**D**) In D37 cortical organoids, 6 organoids were surveyed to count ZO-1-positive apical surfaces and proportions of PAX6-positive NPCs that are organized around the apical surfaces. (**E**) Proportions of PH3-positive cells that are adjacent to ZO-1-positive apical surfaces ("rings"). Data from 53BP1-S25A and S25D were included for comparison. **, $p < 0.01$; ***, $p < 0.001$; ****, $p < 0.0001$; ns, not significant by Welch's $t$ test in (A-C) and two-way ANOVA test in (D). From GSEA, functional terms that are highly enriched in (**F**) up-regulated and (**G**) down-regulated genes in *ATM*-KO vs. WT NPCs. % Match, % of genes in the enriched term that overlap the differentially expressed genes or proteins. Underlying numerical values for figures are found in S1 Data. ATM, ataxia telangiectasia mutated; GSEA, gene set enrichment analysis; KO, knockout; NES, normalized enrichment score; NPC, neural progenitor cell; WT, wild type; 53BP1, p53 binding protein 1.

exhibited higher neuronal maturation (Figs 2A–2C and S5A). These data suggest that *ATM*-KO fastens the phase of immature neuronal differentiation, leading to enhanced neuronal maturation. Finally, we quantified ZO-1-positive ventricular surfaces and the organization of PH3-positive and PAX6-positive cells around ventricular surfaces. The *ATM*-KO ventricular surfaces were similar to WT at D28 (S5B–S5D Fig), but the number was much reduced by D37 (Fig 2D). NPC organization around the ventricular surfaces were similarly organized in D37 (Figs 2D and S5E); however, fewer *ATM*-KO proliferative cells were adjacent to ventricular surfaces (Fig 2E). These data suggest that *ATM*-KO enhances neuronal maturation and cellular disorganization in developing cortical organoids. By D55, *ATM*-KO cortical organoids had similar size distribution as the control (S5F and S5G Fig). Thus, ATM controls neuronal differentiation and cellular organization to form ventricular zone–like regions in cortical organoids.

## ATM safeguards transcriptional and translational programs in differentiating cortical organoids

To investigate the molecular basis of the cellular defects we observed in *ATM*-KO, we performed RNA-seq to compare *ATM*-KO to WT NPCs and D35 cortical organoids derived from WT and *ATM*-KO hESCs. The expression of forebrain markers was similar between WT and *ATM*-KO cortical organoids (and low expression of midbrain and hindbrain markers; S2 Table), suggesting that the *ATM*-KO cortical organoids specified to the forebrain lineage. A false discovery rate (FDR) <0.05 was used to identify differentially expressed genes. Gene set enrichment analysis (GSEA) showed that up-regulated genes in *ATM*-KO NPCs were enriched in forebrain development, axis specification, and metabolic pathways (Figs 2F and S6A), whereas down-regulated genes were enriched in neuronal differentiation, epithelial mesenchymal transition, and tube morphogenesis (Fig 2G). Comparison of transcriptomes profiles of D35 cortical organoids from 8 *ATM*-KO versus 6 WT datasets yielded similar GSEA terms (Fig 3A and 3B). These data suggest that ATM regulates genetic programs related to forebrain development, metabolism, and neuronal differentiation in NPCs and cortical organoids. Dysregulated genetic programs likely contributed to enhanced neuronal maturation and cellular disorganization in *ATM*-KO cortical organoids.

As ATM kinase is crucial for many cell and developmental processes, we aimed to analyze its effect on the proteome and phosphoproteome of differentiating cortical organoids. First, we used multiplexed tandem mass tag-based quantification and 2D liquid chromatography-tandem mass spectrometry (TMT LC-MS/MS) to profile the proteome of WT and *ATM*-KO D35 cortical organoids (S3B Fig, Methods). We quantified 10,895 proteins between 4 WT, 4 *ATM*-KO2, 3 *ATM*-KO3, and 3 *ATM*-KO14 D35 cortical organoid samples by using the criteria of fold change >1.5 and FDR <0.05 (Fig 3C). Consistency between replicate datasets is supported by principal component analysis (S3B Fig). GSEA showed that compared to WT, up-regulated proteins in *ATM*-KO were enriched in terms related to neurotransmission, neuron spine, dendrite, synapse, and axon (S3C Fig), whereas down-regulated proteins were enriched in BMP/TGFβ and WNT signaling, epithelial morphogenesis, and stem cell differentiation (S3D Fig). These data suggest that ATM controls posttranscriptional and translational gene regulation to suppress neuronal function and promote stem cell differentiation, epithelial morphogenesis, and TGFβ and WNT signaling pathways in D35 cortical organoids.

We have observed distinct patterns in the transcriptomics and proteomics data in *ATM*-KO versus WT cortical organoids. Interestingly, while transcriptomic programs related to neuronal differentiation were down-regulated (Fig 3B), proteomic programs related to neuronal function were up-regulated (S3C Fig) in *ATM*-KO versus WT cortical organoids. These differential patterns in transcriptomics and proteomics are likely a consequence of the regulatory

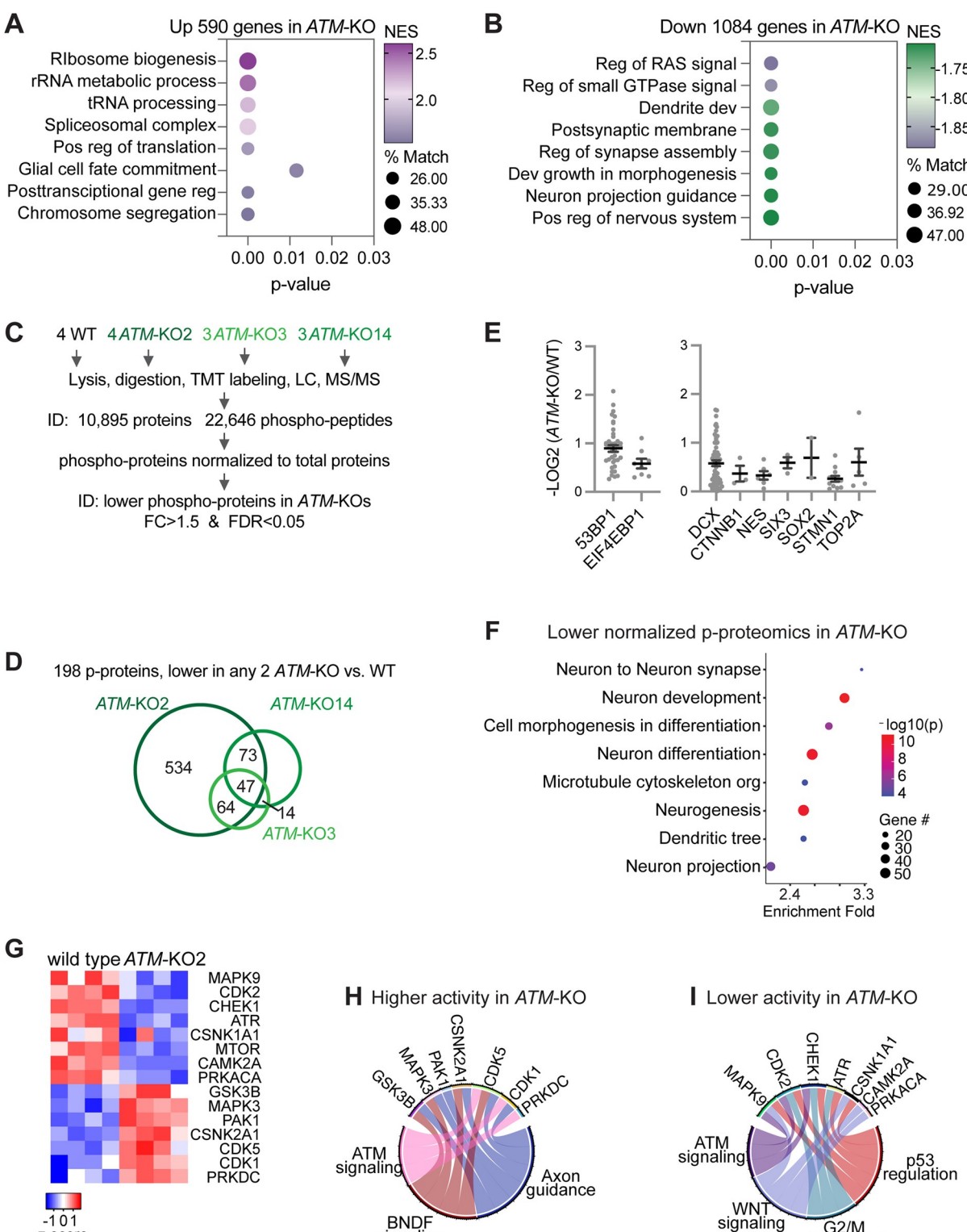

**Fig 3. Transcriptomic and proteomic profiles of WT versus *ATM*-KO cortical organoids.** From GSEA, functional terms that are highly enriched in (**A**) up-regulated and (**B**) down-regulated genes in *ATM*-KO D35 cortical organoids. % Match, % of genes in the enriched term that overlap the differentially expressed genes or proteins. (**C**) Schematic diagram outlining TMT LC-MS/MS profiling of total proteomics and phosphoproteomics of D35 WT and *ATM*-KO cortical organoids. TMT signals from total proteomics were used to normalize those of phosphopeptides. (**D**) Using FC>1.5 and FDR<0.05, 198 phosphoproteins were found to be lower in 2 *ATM*-KO versus WT. (**E**) Normalized

levels of phosphoproteins that have ATM-dependent phosphorylation in D35 cortical organoids. 53BP1 and EIF4EBP1 were known substrates of ATM. The error bars depict the mean and standard error of the mean values, which were calculated based on the normalized levels of each phosphopeptide in the protein. (**F**) Enrichment of proteins with ATM-dependent phosphorylation in specific functional categories. (**G**) Heatmap showing altered activities of kinases between D35 *ATM*-KO2 and WT cortical organoids. Relative changes in kinase activity are shown as row Z-scores. Kinase activity was inferred by IKAP [25] based on normalized substrate phosphorylation levels from phosphor-proteome. The normalization was performed by dividing phosphor-peptide abundance of each protein by corresponding protein abundance [57]. Circos plots showing kinases with inferred (**H**) higher and (**I**) lower activities in D35 *ATM*-KO versus WT cortical organoids and their corresponding enriched pathways. Underlying numerical values for figures are found in S1_Data.xlsx. ATM, ataxia telangiectasia mutated; FC, fold-change; FDR, false discovery rate; GSEA, gene set enrichment analysis; KO, knockout; NES, normalized enrichment score; TMT LC-MS/MS, liquid chromatography-tandem mass spectrometry; WT, wild type; 53BP1, p53 binding protein 1.

role of ATM in multiple cellular processes. It is possible that the higher protein expression related to neuronal functions in *ATM*-KO lead to down-regulation of transcriptional expression of neuronal differentiation programs. This would suggest that the dysregulated transcriptomic and proteomic programs in ATM-KO cortical organoids are interconnected and result from the complex interplay of ATM's regulation of various cellular pathways. These findings shed light on the intricate role of ATM in coordinating gene expression and protein levels, influencing neuronal differentiation and function in cortical organoids.

## ATM-dependent phosphorylation controls signaling pathways for neurogenesis, stem cell differentiation, and morphogenesis in cortical organoids

To investigate how ATM exerts its modulatory control during cortical organoid formation, we performed phosphoproteomics analysis of WT and *ATM*-KO cortical organoids. Using TMT LC-MS/MS, we quantified 22,646 phosphopeptides and normalized their abundance based on the protein abundance measured in the total proteomics analysis. A comparison between WT and *ATM*-KO lines revealed that 198 proteins had consistently lower levels of phosphorylation in at least 2 of the 3 *ATM*-KO lines (log2(fold change >1.5) and FDR <0.05; Fig 3D and S3 Table). Among these proteins, 53BP1 and EIF4EBP1 were known substrates of ATM (Fig 3E) [15,16,22,23], validating the approach to identify putative ATM substrates in cortical organoids. However, it is essential to note that this approach does not distinguish between direct and indirect effects, and, therefore, some of the identified proteins could be phosphorylated by protein kinases that require ATM for their activity. Notably, many ATM-dependent phosphorylated proteins were found to be key neurodevelopmental regulators (Fig 3E) and enriched in functions related to neurodevelopment, neurogenesis, cell morphogenesis, and cytoskeleton (Fig 3F). These findings suggest that ATM plays a critical role in regulating the phosphorylation of proteins involved in essential processes for neurodevelopment and neuronal function in cortical organoids.

We further explored the effects of ATM by identifying protein kinases that had ATM-dependent phosphorylation. We used the IKAP machine learning algorithm [24] to analyze substrates (inferred from literature curation) and deduce the activities of those kinases. For example, in *ATM*-KO compared to WT, we found reduced phosphorylation of proteins related to MAPK9 activities, such as DCX, MAPT, and NFATC4 (S7A Fig and S4 Table) [24]. On the other hand, we found higher phosphorylation of proteins related to CDK5 activities, including ADD2, ADD3, DCX, DNM1L, DPYSL3, MAPT, and SRC (S7B Fig and S4 Table) [24]. In *ATM*-KO, we inferred lower activities in MAPK9, CDK2, CHEK1, ATR, CSNK1A1, MTOR, CAMK2A, and PRKACA (Figs 3G and S7C), with enriched functions in ATM signaling, BNDF signaling, and axon guidance (Figs 3G, 3H, and S7C). On the other hand, we inferred higher activities in GSK3B, MAPK3, PAK1, CSNK2A1, CDK5, CDK1, and PRKDC (Figs 3G, 3I, and S7C), with enriched function in ATM signaling, WNT signaling, G2/M checkpoint,

and p53 regulation in *ATM*-KO (Fig 3H and 3I). *ATM* KO leads to both lower and higher activities of kinases in ATM signaling. Additionally, some of the altered kinase activities could be secondary to *ATM*-KO, as CHEK1, ATR, and PRKDC were known substrates of ATM [23,25]. Overall, these data suggest that the activities of kinases related to ATM signaling, BNDF signaling, WNT signaling, G2/M checkpoint, and p53 regulation became dysregulated in *ATM*-KO D35 cortical organoids.

We thus conclude that ATM plays a crucial role in controlling key neurodevelopmental regulators. The dysregulated phosphorylation and activities of these regulators disrupt the normal transcriptomic program responsible for neuronal differentiation, leading to higher proteomic programs associated with neuronal function. As a consequence, the dysregulated programs in *ATM*-KO cortical organoids are likely responsible for the observed defects in neurogenesis and morphogenesis (formation of ventricular zone–like regions). These findings provide valuable insights into the role of ATM in neurodevelopment and shed light on potential molecular mechanisms underlying neurological disorders associated with ATM dysfunction.

## Phosphorylation of 53BP1-S25 coordinates NPC proliferation and neuronal differentiation

We next examined ATM-dependent phosphorylation of 53BP1-S25. To specifically investigate the functional significance of 53BP1-pS25, we used the CRISPR-Cas9 system to mutate the endogenous 53BP1 serine 25 to alanine (S25A) or aspartic acid (S25D) (Figs 4A and S7D, Methods). The alanine substitution precludes phosphorylation, whereas aspartic acid is chemically similar to phosphoserine [26]. We generated 4 53BP1-S25A hESC lines (34–3, 34–4, 79–1, and 79–3) and 4 53BP1-S25D hESC lines (14–3, 14–15, 14–19, and 17). The total levels of 53BP1 were similar in WT, 53BP1-S25A, and 53BP1-S25D NPCs, and we did not detect pS25 in 53BP1-S25A NPCs, as expected (S1D and S7E Figs). Control, 53BP1-S25A, and 53BP1-S25D hESC lines displayed similar transcriptomic profiles and pluripotency marker expression (S2C, S7F, and S8A Figs), suggesting that 53BP1-S25A and 53BP1-S25D do not affect hESC self-renewal.

ATM is required for the phosphorylation of many neurodevelopmental regulators (Fig 3F). As the role of 53BP1-S25 beyond DNA damage repair is not known, we seek to analyze its role in human cortical development. We differentiated control WT, 53BP1-S25A, and 53BP1-S25D hESCs into cortical organoids (Fig 1E, Methods). The 53BP1-S25A and 53BP1-S25D D35 cortical organoids displayed smaller sizes compared to WT controls (Fig 4B and 4C), suggesting that phosphorylation at S25 is essential for cortical organoid growth and development. Analysis of the ventricular zone–like regions showed 53BP1-S25A and 53BP1-S25D are significantly smaller than those in WT (Fig 4D-4F). Fewer cells were positive for KI67 (proliferation marker) or phosphorylated-serine 10 histone H3 (mitotic chromatin marker) (S8B–S8D Fig). Examination of endogenous DNA damage and cell death, assessed by γH2AX and cleaved-caspase 3, respectively, did not reveal significant differences between 53BP1-S25A, S25D, and WT (S9 Fig). To explore the developmental timing of the cellular phenotypes, we quantified KI67, NPC marker PAX6, and neuronal marker CTIP2 in D14, D21, D28, and D35 cortical organoids. At D28, 53BP1-S25A and S25D cortical organoids had significantly lower cell proliferation and higher neuronal differentiation (Fig 5). Quantification of the tight junction protein ZO-1 showed significantly fewer ZO-1-positive ventricular surfaces in D28 53BP1-S25A and S25D cortical organoids compared to WT (S10 Fig). For the ventricles that did form in D28 53BP1-S25A and S25D, their surface areas did not significantly differ from those of WT (S10B Fig). These data suggest that lower ventricle formation, lower cell proliferation, and higher neuronal differentiation contributed to the depletion of progenitor pools and smaller cortical organoids in 53BP1-S25A and S25D.

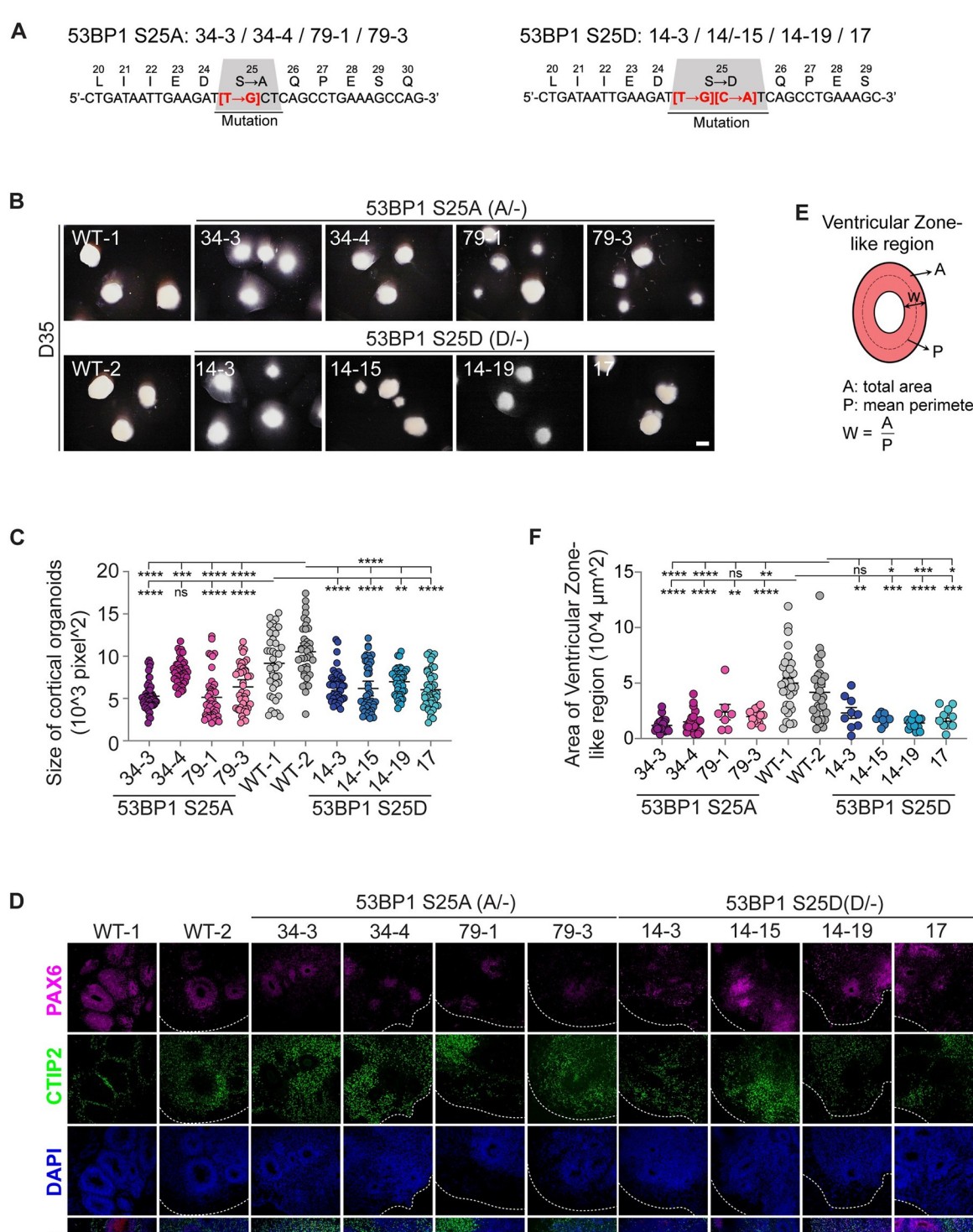

**Fig 4. 53BP1-pS25 is required for the differentiation of cortical organoids.** (**A**) In the endogenous *53BP1* locus, the codon TCT encoding serine-25 in was mutated to GCT and GAT encoding alanine and glutamate, respectively. (**B**) Bright-field images of cortical organoids formed by 4 53BP1-S25A lines, 4 53BP1-S25D lines, and 2 WT control at day 35 of differentiation. Bar, 1.5 mm. At day 35 of differentiation, the (**C**) organoid size and (**F**) area of ventricular zone–like region were compared between groups. Data points represent single organoids. The mean ± SEM values were compared by one-way ANOVA with Dunnett's multiple comparisons test to yield ****,

***, **, *, and ns indicating $p < 0.0001$, 0.001, 0.01, 0.05, and not significant, respectively. $n = 39$–47 organoids/group for (**C**) and 15–33 organoids/group for (**F**). (**D**) Immunofluorescence of PAX6 and CTIP2 in cryosections of cortical organoids at day 35 of differentiation. Bar, 100 μm. (**E**) Illustration of ventricular zone–like areas in cortical organoids. Underlying numerical values for figures are found in S1 Data. WT, wild type; 53BP1, p53 binding protein 1; 53BP1-pS25, 53BP1 phosphorylated at serine 25.

At D55, the 53BP1-S25A and S25D cortical organoids remained significantly smaller than WT (S12 Fig and S5 and S6 Tables). These data suggest that the cell biological effects of the S25A and S25D mutations were similar, despite the aspartic acid mutation (S25D) being chemically similar to phosphoserine, which is the phosphorylated form of S25. The S25D mutation may act as an inhibitory mimic of phosphorylation, akin to the S25A mutation. Consequently, the absence of S25 phosphorylation impacts NPC proliferation and overall cortical organoid growth.

## Phosphorylation of 53BP1-S25 modulates the expression of genetic programs for neuronal differentiation and function

Using RNA-seq, we examined the transcriptomes of WT (6 samples), 53BP1-S25A (8 samples), and 53BP1-S25D (8 samples) D35 cortical organoids. We analyzed expressed genes with counts per million values >1 and observed few differences in gene expression between 53BP1-S25A and 53BP1-S25D D35 cortical organoids, using FDR <0.05 (Fig 6A). When comparing the transcriptomes of 53BP1-S25A and 53BP1-S25D organoids to WT, there were high concordant changes in gene expression, with over 87% of differentially expressed genes in 53BP1-S25A also being altered in 53BP1-S25D (Figs 6B and S12D). However, 53BP1-S25D disrupted the expression of 2- to 3-fold more genes than 53BP1-S25A, suggesting a gain-of-function effect for the 53BP1-S25D mutation. To explore this further, we performed GSEA and found that the top terms enriched in the up-regulated genes of 53BP1-S25A and S25D organoids were highly overlapping (Fig 6C). In contrast, there was low overlap of the top terms in the down-regulated genes in 53BP1-S25A versus WT and 53BP1-S25D versus WT (S12E Fig). Both mutations led to the up-regulation of genes related to synapse, axon, and neurotransmitter functions, suggesting a shared effect on enhancing neuronal function (Fig 6C). The S25D mutation specifically up-regulated more genes involved in neuronal function compared to S25A, indicating a stronger impact on this aspect of gene regulation (Fig 6C and 6D). These findings highlight the significance of the S25 phosphorylation site in 53BP1 for the regulation of genes involved in neuronal function and support that the S25D mutation results in a gain-of-function effect, leading to more pronounced changes in gene expression related to neuronal processes.

It remained unclear whether the higher expression of neuronal genetic programs in the 53BP1 mutants occurred in NPCs or neurons. Therefore, we compared the transcriptomes of 53BP1-S25A and S25D to WT NPCs, which had similar expression of NPC markers PAX6 and NES (S12A Fig). Up-regulated genetic programs in 53BP1-S25A and S25D NPCs shared categories such as translation control and ribosome (S12B and S12C Fig), whereas 53BP1-S25A NPCs also up-regulated cell cycle control and chromosome segregation (S12C Fig). Surprisingly, down-regulated genetic programs in 53BP1-S25A and S25D NPCs were highly enriched in neuronal differentiation (S12D and S12E Fig). The down-regulated genetic programs in NPCs are similar to neuronal programs that became up-regulated in 53BP1-S25A and S25D versus WT cortical organoids. These data suggest that 53BP1-S25 phosphorylation promotes the appropriate expression of neurogenic programs in NPCs and modulates the expression of the same programs in differentiating neurons in cortical organoids.

To dig deeper into analyses, we compared our data with previously published transcriptomic data that compared *53BP1*-KO and WT cortical organoids, which support a

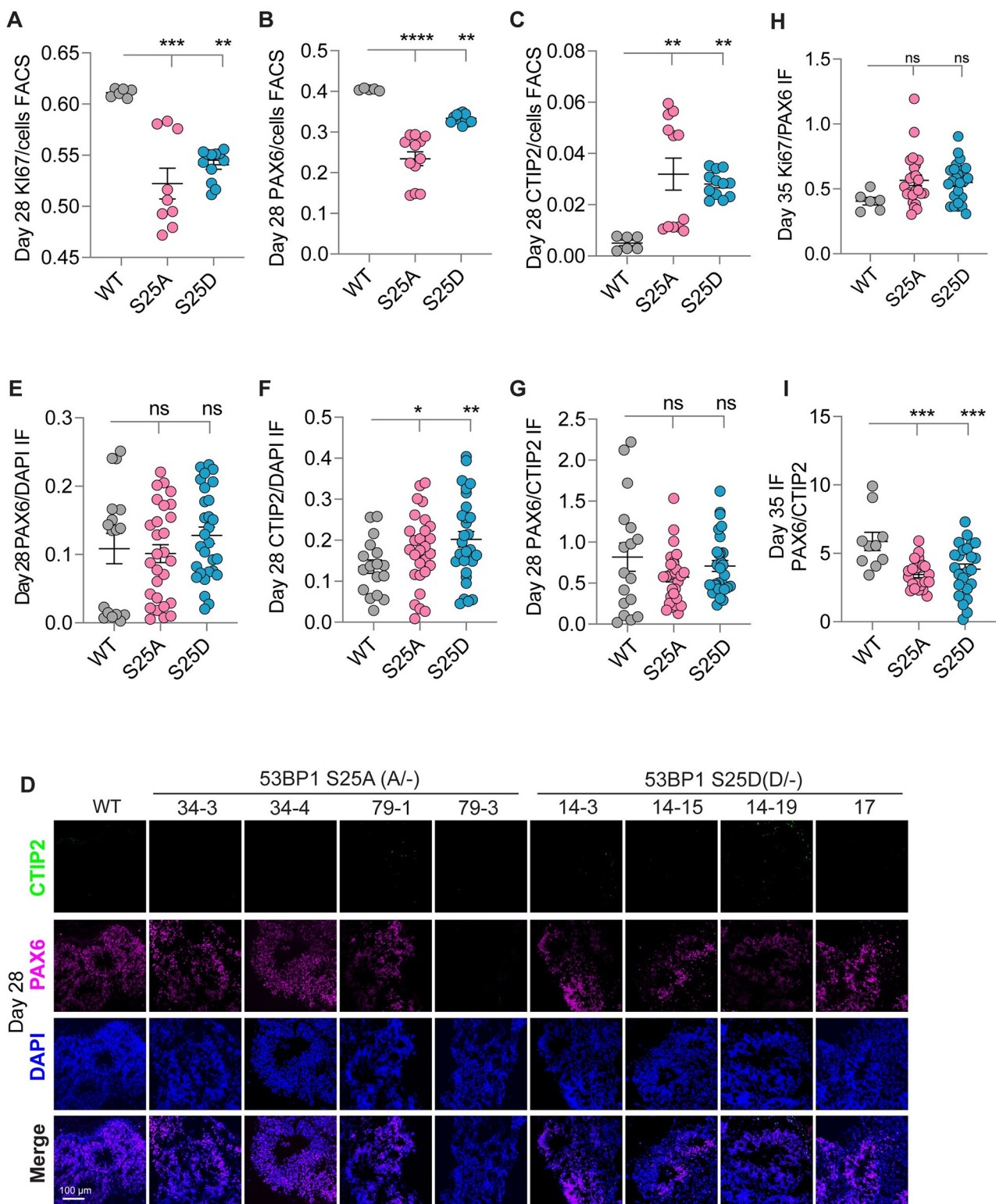

**Fig 5. 53BP1-S25A and S25D lower cell proliferation in cortical organoids.** FACS quantified ratios of (**A**) KI67, (**B**) PAX6, and (**C**) CTIP2 to total cells in D28 cortical organoids. (**D**) Immunofluorescence of PAX6 and CTIP2 in D28 cortical organoids. Bar, 100 μm. Quantification of immunofluorescence signals of (**E**) PAX6/DAPI, (**F**) CTIP2/DAPI, and (**G**) PAX6/CTIP2 in D28 cortical organoids. Each data point represents quantification of cells in 1 cortical organoid. Quantification of (**H**) KI67/PAX6 and (**I**) PAX6/CTIP2 ratios in immunofluorescence of D35 cortical organoids. Each data point represents quantification of cells in 1 cortical organoid. *, $p < 0.05$; **, $p < 0.01$; ***, $p < 0.001$; ****, $p < 0.0001$; ns, not significant by two-way ANOVA test. Underlying numerical values for figures are found in S1 Data.

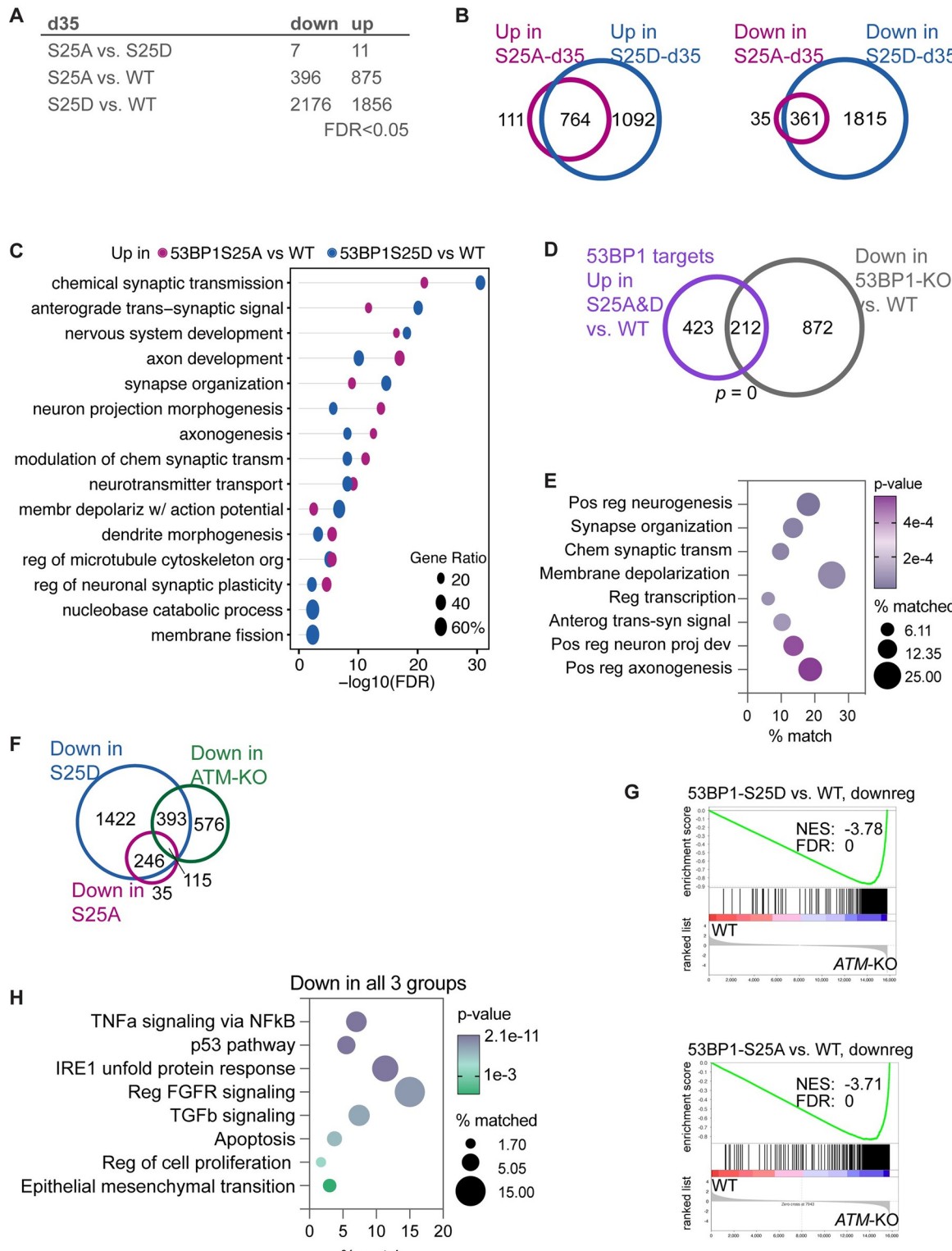

**Fig 6. 53BP1-S25 phosphorylation enforce the appropriate expression of genetic programs for cortical organoid differentiation.**
(**A**) Number of differentially expressed genes identified by pairwise comparisons at FDR <0.05. At day 35 of differentiation, 53BP1-S25A and S25D cortical organoids are molecularly similar. (**B**) Differentially expressed genes in 53BP1-S25D versus WT overlap 87% (764/ 875) and 91% (361/396) of those in 53BP1-S25A versus WT. (**C**) Extensive overlap of up-regulated GSEA terms between 53BP1-S25A versus WT and 53BP1-S25D versus WT. Most terms relate to axon, synapse, and neurotransmitter. (**D**) Of 53BP1 target genes up-

regulated by S25A and S25D, 212 genes require WT 53BP1 for expression in cortical organoids. (**E**) The 212 genes are enriched in functions related to transcriptional regulation, neuron projection, axonogenesis, synapse, neurotransmitter synthesis and transport, and membrane depolarization. (**F**) Venn diagrams depict high overlaps between down-regulated genes in all 3 groups of mutant versus WT pairwise comparisons. (**G**) GSEA graphs showed that down-regulated genes in 53BP1-S25A or S25D vs. WT had significant enrichment in down-regulated genes of ATM-KO vs. WT cortical organoids. *P* values were calculated by the hypergeometric test, assuming normal data distribution. (**H**) GSEA terms of the 115 genes that were down-regulated in all 3 groups (versus WT) revealed the genetic programs copromoted by ATM and 53BP1-pS25. ATM, ataxia telangiectasia mutated; FDR, false discovery rate; GSEA, gene set enrichment analysis; KO, knockout; WT, wild type; 53BP1, p53 binding protein 1.

requirement of 53BP1 for activating neurogenic genes [6]. We observed that gene categories up-regulated by 53BP1-S25A and S25D were similar to those down-regulated in *53BP1*-KO cortical organoids. This was a significant overlap of 212 genes up-regulated by 53BP1-S25A and 53BP1-S25D with 53BP1-bound target genes that were down-regulated in *53BP1*-KO versus WT (*p* = 0 by empirical estimation; Fig 6D). The 212 genes were enriched in functions related to regulation of transcription, neurogenesis, neuronal projection, axonogenesis, synapse organization, and membrane depolarization (Fig 6E). This suggests that the expression of these genes is dependent on and up-regulated by 53BP1 phosphorylated at S25 in cortical organoids.

We next examined how transcriptomic changes in 53BP1-S25A and S25D compared to those in *ATM*-KO cortical organoids. We observed little overlap between the down-regulated genes in *ATM*-KO and the up-regulated genes in 53BP1-S25A and 53BP1-S25D. In contrast, we observed a greater overlap in concordant gene expression changes in *ATM*-KO, 53BP1-S25A, and 53BP1-S25D versus WT (Figs 6F, S12F, and S12G). GSEA showed a significant enrichment of concordantly differentially expressed genes among *ATM*-KO, 53BP1-S25A, and 53BP1-S25D versus WT (Figs 6G and S13A-S13C). Notably, all 3 mutant types shared down-regulated genes that were enriched functions related to TNFα signaling via NFκB, p53 pathway, IRE1-mediated unfolded protein response, FGFR signaling, TGFβ signaling, apoptosis, regulation of cell proliferation, and epithelial mesenchymal transition (Fig 6G). These data suggest that both ATM and 53BP1-pS25 promote the expression of these genes. From these findings, we can infer that ATM likely promotes the expression of these genes via phosphorylating 53BP1 at S25 in D35 cortical organoids. This suggests that ATM and 53BP1 may function together in a coordinated manner to regulate the expression of genes involved in critical signaling pathways and cellular processes during cortical development.

## 53BP1-S25A and S25D predominantly alter the expression of 53BP1 target genes

To obtain further mechanistic insights into the role of 53BP1 in controlling gene expression, we reanalyzed 53BP1 ChIP-seq data (using 2 separate anti-53BP1 antibodies) in WT NPCs [6]. Using SICER [27] and MACS2 [28] with a criterion of FDR <0.05, we identified 37,519 targets bound by 53BP1. About 41% of these 53BP1 targets localize to promoter regions, suggesting a transcriptional regulatory role of 53BP1 (S14D Fig). Remarkably, more than 82% of the differentially expressed genes in 53BP1-S25A and 53BP1-S25D D35 cortical organoids were found to be targets bound by 53BP1 (Figs 7A, S13E, and S13F). 53BP1 target genes with increased transcript levels in the mutant organoids were highly enriched in neuronal development, axonogenesis, neuron projection, synapse organization, and neurotransmitter transport, transmission, and signaling (Fig 7B). On the other hand, 53BP1 targets with reduced transcript levels in the mutant organoids were enriched in IRE1-mediated unfolded protein response, cellular response to stress, iron import, and apoptosis regulation (S13G Fig). Of note, genes involved in IRE1-mediated unfolded protein response and apoptosis regulation showed reduced

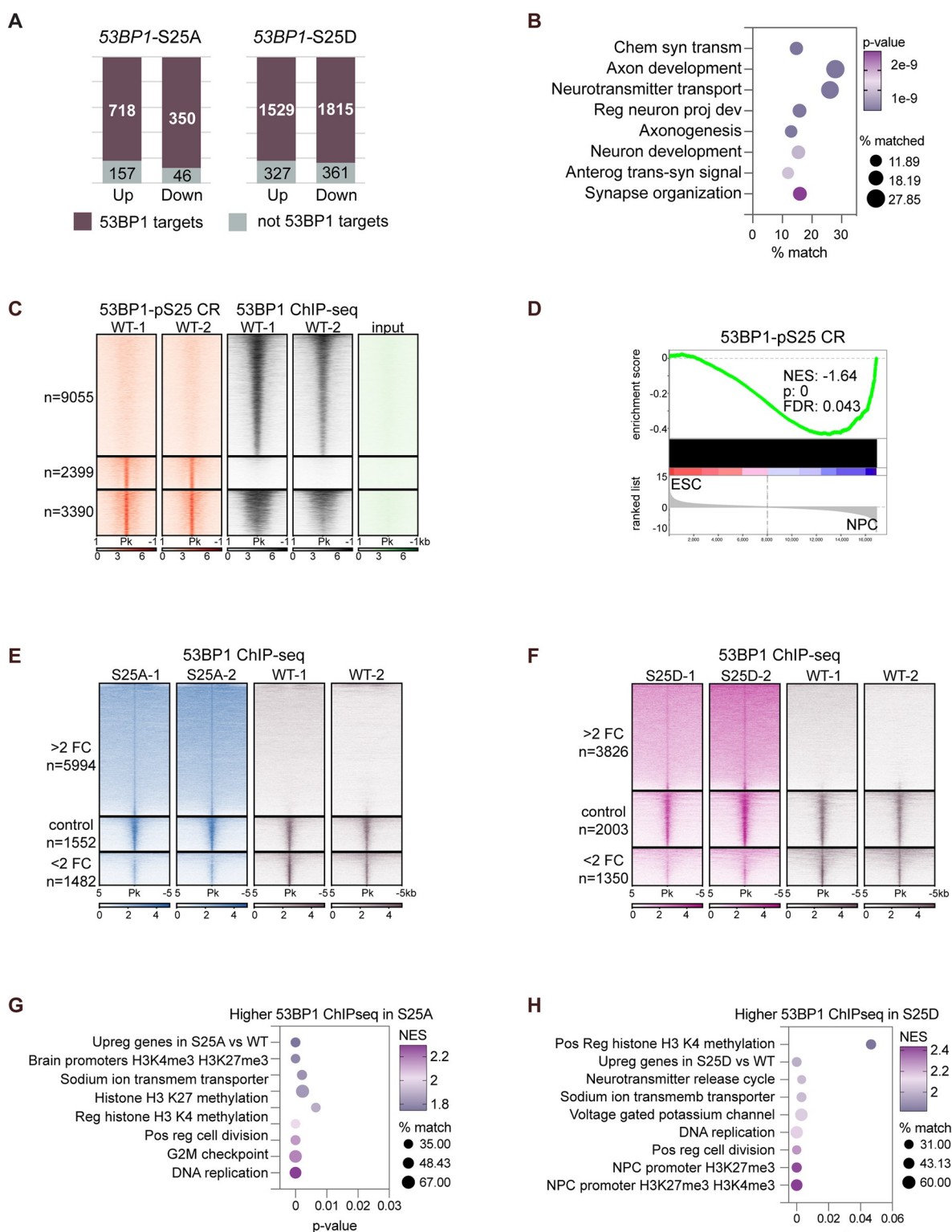

**Fig 7. 53BP1-pS25 positively and negatively regulate 53BP1 target genes.** (**A**) More than 82% of differentially expressed genes in 53BP1-S25A or S25D versus WT are chromatin targets bound by 53BP1 in WT NPCs. (**B**) 53BP1-S25A and S25D up-regulate 53BP1 targets that are involved in neuron development and projection, axonogenesis, synapse, and neurotransmitter synthesis and transport. (**C**) Heatmaps aligning peaks with 53BP1-pS25 CUT&RUN and 53BP1 ChIP-seq signals in WT NPCs. Input track was included as a negative control. n = numbers of peaks with differential and overlapped bindings. Criteria of FC>2 and *p* < 0.05 were used for comparison. (**D**)

GSEA graph of 53BP1-pS25 CUT&RUN signals in genes that were lower in ESCs vs. NPCs, which were up-regulated in NPCs. *P* values were calculated by the hypergeometric test, assuming normal data distribution. Heatmaps aligning peaks with significantly different 53BP1 ChIP-seq signals in (**E**) 53BP1-S25A vs. WT and (**F**) 53BP1-S25D vs. WT, using the criterion of FC>2 and *p* < 0.05. Control peaks are those, after voom normalization, showed the least changes and served as semi-independent validation of differential ChIP-seq analysis. Bubble graphs present top enriched categories of genes that had significantly higher 53BP1 ChIP-seq in (**G**) 53BP1-S25A vs. WT and (**H**) 53BP1-S25D vs. WT. ESC, embryonic stem cell; FC, fold-change; GSEA, gene set enrichment analysis; NPC, neural progenitor cell; WT, wild type; 53BP1, p53 binding protein 1; 53BP1-pS25, 53BP1 phosphorylated at serine 25.

expression upon loss of ATM or mutation of 53BP1-S25 and were identified as direct targets of 53BP1 in NPCs. This suggests that ATM-mediated phosphorylation of 53BP1-S25 directly promotes the expression of these genes to maintain NPCs during formation of cortical organoids.

We wanted to test whether ATM alters 53BP1 binding, considering ATM is required for 53BP1-pS25 in D35 cortical organoids (Fig 1F) and NPCs (S2B Fig). A comparison of 53BP1 ChIP-seq in WT and *ATM*-KO NPCs showed that *ATM*-KO altered 53BP1 binding to chromatin (S14A-S14C Fig). *ATM*-KO reduced 53BP1 binding at specific sites, with 96.3% of these sites being promoters (S14C Fig). To explore the impact of 53BP1-pS25, we performed CUT&RUN in 2 separate WT NPC lines. Our analysis revealed that 67.1% of 53BP1-pS25 targets localize to promoter regions, suggesting a transcriptional regulatory role (S14D Fig). Under the criteria of fold-change >2 and *p* < 0.05, 58.6% (3,390/5,789) of 53BP1-pS25 targets overlapped with 53BP1 targets (Fig 7C); the nonoverlapped sites may be attributed to differences in ChIP versus CUT&RUN procedures and the accessibility of 53BP1 versus 53BP1-pS25 antibodies. 53BP1-pS25 targets were significantly enriched in 414 up-regulated genes in NPCs versus ESCs (Fig 7D), suggesting a role of 53BP1-pS25 in promoting their expression in NPCs. Genes having overlapped 53BP1 ChIP-seq and 53BP1-pS25 CUT&RUN signals were enriched in chromatin remodeling, DNA metabolism, RNA splicing, translation, transcription, cell cycle, and neuron development (S14E Fig). These data suggest that ATM can alter 53BP1 binding and that 53BP1-pS25 is enriched in the promoters of genes regulating cellular processes and neurodevelopment.

## Phosphorylation of 53BP1-S25 controls the localization of 53BP1 to chromatin for gene regulation

To investigate the impact of 53BP1-S25 on the genomic distribution of 53BP1, we performed ChIP-seq in 53BP1-WT, S25A, and S25D NPCs. Two independent NPC lines were used for each group, and the ChIP-seq data were subjected to principal component analysis, which showed high consistency between the replicate dataset (S8A Fig). We used SICER [27] and MACS2 [28] with the criteria of fold-change >2 and *p* < 0.05 to perform pairwise comparisons of the merged datasets from 53BP1-WT, S25A, and S25D ChIP-seq experiments. The pairwise comparisons identified thousands of 53BP1-bound regions that were significantly different between 53BP1-WT, S25A, and S25D. Notably, the regions that significantly gained binding in 53BP1-S25A or S25D versus WT were highly enriched at promoters (within 2 kb of transcription start sites), constituting 82% and 71.1%, respectively (S8B and S8C Fig). In contrast, the regions that significantly lost binding in 53BP1-S25A or S25D versus WT were not as enriched at promoters, constituting 32.6% and 33%, respectively (S8B and S8C Fig). We generated heatmaps to visualize the genomic regions with significantly different 53BP1 binding intensity (compared against control regions). The heatmaps confirmed consistent changes in 53BP1 binding patterns between 53BP1-S25A and S25D versus WT, and between 53BP1-S25A versus S25D (Figs 5C, 5D, and S8D). These data support that 53BP1-S25 and its phosphorylation control the genomic distribution of 53BP1 on chromatin.

We next set out to examine the correlation between changes in 53BP1 distribution on chromatin and changes in gene expression in 53BP1-WT, S25A, and S25D cortical organoids. We performed GSEA and made some notable observations. Firstly, regions that gained 53BP1 binding in 53BP1-S25A or S25D cortical organoids, as compared to WT, were enriched with up-regulated genes (Fig 5E and 5F). Similarly, regions that had lower 53BP1 binding were enriched with down-regulated genes in 53BP1-S25A or S25D versus WT (S8E and S8F Fig). These results suggest that the 53BP1-S25A or S25D mutation directly influences 53BP1 binding and gene expression and subsequently regulates gene expression, particularly at promoters where higher 53BP1 binding leads to higher gene expression.

Interestingly, genes that lost 53BP1-S25A or S25D protein binding had minimal overlap in GSEA terms, except promoters occupied with H3K4me3 and regulation of epithelial-mesenchymal transition (S8E and S8F Fig). In contrast, the genes that gained 53BP1-S25A or S25D protein binding were enriched with promoters marked by bivalent histone marks (H3K4me3 and H3K27me3) or occupied by H3K27me3 alone [29] (Fig 5E and 5F), suggesting that 53BP1-S25D or S25A proteins preferentially bind to these promoters and subsequently up-regulate gene expression. Moreover, genes that gained 53BP1-S25A or S25D binding shared common functions related to sodium ion transmembrane transporter, DNA replication, positive regulation of cell division, and regulation of histone H3K4 methylation (Fig 5E and 5F). This suggests that despite the 1,187 regions showing different 53BP1 bindings between 53BP1-S25A and S25D (S8D Fig), both mutations impact genes involved in neuronal functions and cell proliferation. Altogether, these findings show that 53BP1-S25A and S25D mutations have a direct impact on 53BP1 binding to chromatin and subsequently affecting gene regulation. We propose that 53BP1-pS25 likely inhibits 53BP1 binding to promoters associated with bivalent and H3K27me3-occupied promoters. This inhibition may lead to the reduced expression of genes involved in the regulation of H3K4me3, neuronal functions, and cell proliferation.

## Molecular regulation of ATM and 53BP1-pS25 during neural differentiation

We next tried to identify a regulation of ATM, whose protein levels increased in NPCs (Fig 1D). This led us to test whether and how inhibitors of TGFb, WNT, and HH signaling control protein levels of ATM, 53BP1, and pS25-53BP1 by removing one inhibitor at a time from the cortical organoid differentiation media (S16A Fig). As we could not successfully identify physical presence of ATM at promoters, WB analysis is most apt to study ATM level and activity. By day 4 of neural differentiation, although 53BP1 protein levels were reduced by the withdrawal of SB431542 (TGFβ inhibitor) or IWR1-endo (WNT inhibitor), pS25-53BP1 was not altered (S16B and S16C Fig). By day 10 of neural differentiation, the withdrawal of cyclopamine (HH inhibitor) reduced pS25-53BP1 level (but not ATM or 53BP1 proteins; S16D and S16E Fig). These signaling pathways may affect pS25-53BP1 or ATM activities during neural differentiation.

Next, we tested whether another DNA damage response factor, apart from ATM, influences 53BP1-pS25. RNF168 plays a central role in the γH2AX-MDC1-RNF8-RNF168-H2AK15ub axis, which governs the binding of 53BP1 to chromatin with DNA damage [30]. We generated *RNF168*-KO hESC clone 44, which maintained pluripotency and genome integrity (S17A-S17D Fig and S1 Table). *RNF168*-KO hESCs were differentiated to NPCs, which expressed NPC markers similar to WT NPCs (S17E Fig). RNA-seq analysis comparing 2 datasets each from *RNF168*-KO44 and WT NPCs revealed that up-regulated genes were enriched in neuronal differentiation, translation and ribosome, and cell cycle transition (S17F Fig), while down-regulated genes were enriched in cilium movement, H3K27me3 targets,

H3K4me3 targets, astrocyte markers, signaling pathways, and positive regulation of NPC proliferation (S17G Fig). We performed 53BP1-pS25 CUT&RUN and showed that *RNF168*-KO disrupted 53BP1-pS25 localization on chromatin (S17H Fig). *RNF168*-KO increased 53BP1-pS25 levels at genes enriched in neuronal differentiation, cell morphogenesis, and stem cell maintenance, whereas *RNF168*-KO decreased 53BP1-pS25 levels at genes enriched in cell cycle transition, signaling receptor regulation, anterior-posterior patterning, and transcription activator (S17I Fig). The altered 53BP1-pS25 localization correlated with differential gene expression in RNF168-KO versus WT NPCs (S17J Fig). Altogether, these data suggest that DNA damage signaling regulates 53BP1 binding to chromatin, affecting genetic programs related to signaling pathways, protein translation, and NPC proliferation and differentiation."

## Discussion

In our study, we made significant discoveries regarding the role of ATM and 53BP1-pS25 in controlling gene expression during the differentiation of hESCs into cortical organoids. We revealed that ATM exerts a strong influence over various aspects of gene regulation, including transcriptional, posttranscriptional, and translational control. While our in vitro model may not fully recapitulate neurodevelopment in vivo, it provides valuable insights into corticogenesis. We have shown that neural differentiation promotes ATM protein levels, and ATM-dependent phosphorylation predominantly impacts factors involved in neurogenesis, neuronal differentiation, cell morphogenesis, and microtubule cytoskeleton. Dysregulation of these processes led to the cellular defects in *ATM*-KO cortical organoids. We showed that key signaling pathways may affect ATM during neural induction. The activity of ATM can be regulated by DNA damage response, reactive oxygen species, hypothxia, hypothermia, and phosphatase WIP1 [31–33]. The exact clarification of mechanisms promoting ATM activities, especially in directing its kinase activity at specific promoters, is beyond the scope of this study. Additionally, we have identified kinases involved in ATM, BDNF, and WNT signaling, G2/M checkpoint, and p53 regulation as being influenced by ATM-dependent phosphorylation during cortical organoid differentiation. These molecular pathways may function in diseases associated with ATM, including ataxia telangiectasia [34–36].

We recognized the diverse effects of ATM and decided to focus our studies on 53BP1-pS25, a phosphorylation event dependent on ATM. We found that 53BP1-pS25 regulates genetic programs including signaling pathways, p53 regulation, apoptosis, and cell proliferation. To understand the mechanisms underlying 53BP1's involvement in gene regulation, we built a model that incorporates current knowledge about 53BP1 functions in the DNA damage response. We propose that ATM phosphorylates H2AX at transcription start sites [13,14], facilitating the recruitment of 53BP1 and subsequent phosphorylation of 53BP1-S25. RNF168, key to DNA damage response signaling [30], also regulates 53BP1-pS25 on chromatin and genetic programs crucial to neural differentiation. Phosphorylation of 53BP1-S25 inhibits the recruitment of 53BP1 to bivalent or H3K27me3-occupied promoters for suppressing the expression of genes involved in the regulation of H3K4me3, neuronal functions, and cell proliferation. The fidelity of gene expression in cortical brain organoids requires dynamic changes in the phosphorylation of 53BP1-S25. This process is likely to involve the interactions of 53BP1 with other proteins, including RIF1, SCAI, and UTX [6,11,12]. These interactors have known roles in chromatin alterations and gene regulation. Notably, UTX is an H3K27me3 demethylase that can modify bivalent or H3K27me3-occupied promoters and has been shown to partner with 53BP1 to promote neurogenesis in humans but not in mice [6]. Given our findings, we propose that 53BP1-pS25 may influence the activities of 53BP1–UTX at bivalent or

H3K27me3-occupied promoters, thus modulating gene expression and contributing to the timing of neuronal differentiation.

Our studies have uncovered the remarkable role of ATM–53BP1 in regulating neurodevelopmental programs. Its impact is multifaceted. Firstly, ATM–53BP1 plays a crucial role in maintaining NPCs and controlling the size of cortical organoids. Secondly, ATM–53BP1 is involved in driving and modulating programs related to synapse formation, axon development, and neurotransmitter regulation, processes fundamental for establishing neuronal networks and communication within the brain. Thirdly, our findings reveal a temporal component in the regulation of neurodevelopmental programs by ATM–53BP1. As cortical organoids progress in differentiation, there is a temporal regulation of neuronal differentiation and function. This switch involves ATM and the 53BP1-pS25 dynamics to specifically control genes associated with synapse, axon, and neurotransmitter, which are crucial to cognition. In the future, elucidation of this mechanism will provide valuable insights into the molecular control of corticogenesis. Beyond 53BP1, ATM-dependent phosphorylation likely controls many other key neurodevelopmental regulators. Future studies of how ATM selects substrates to exert its multiple influences will significantly advance our understanding of the epigenetic programming underlying human neurodevelopment.

## Materials and methods

### Buffers

PBS: 137 mM NaCl, 2.7 mM KCl, 10 mM Na2HPO4, 1.8 mM KH2PO4 (pH 7.4). PBST: PBS with 0.1% Triton X-100. HEPM: 25 mM HEPES (pH 6.9), 10 mM EGTA, 60 mM PIPES, 2 mM MgCl2. Immunofluorescence blocking solution: 1/3 Blocker Casein (Thermo Fisher Scientific), 2/3 HEPM with 0.05% TX-100. Buffer A: 10 mM HEPES (pH 7.9), 10 mM KCl, 1.5 mM MgCl2, 0.34 M sucrose, 10% glycerol. Buffer B: 3 mM EDTA, 0.2 mM EGTA. Buffer D: 400 mM KCl, 20 mM HEPES, 0.2 mM EDTA, 20% glycerol. ChIP lysis buffer 3: 10 mM Tris-HCl (pH 8.0), 100 mM NaCl, 1 mM EDTA, 0.5 mM EGTA, 0.1% sodium deoxycholate, 0.5% N-Lauroylsarcosine. ChIP wash buffer: 50 mM HEPES (pH 7.5), 500 mM LiCl, 1 mM EDTA, 1% NP-40, 0.7% Na-deoxycholate. ChIP elution buffer: 50 mM Tris-HCl (pH 8.0), 10 mM EDTA, 1% SDS. CUT&RUN Wash buffer: 20 mM HEPES (pH 7.5), 150 mM NaCl, 0.5 mM spermidine, protease inhibitor cocktail (Sigma-Aldrich 11873580001). CUT&RUN Binding buffer: 20 mM HEPES-KOH (pH 7.9), 10 mM KCl, 1 mM CaCl2, 1 mM MnCl2. CUT&RUN Digitonin buffer: CUT&RUN Wash buffer with 0.01% digitonin. CUT&RUN Antibody buffer: CUT&RUN Digitonin buffer with 2 mM EDTA. CUT&RUN 2X Stop buffer: 340 mM NaCl, 20 mM EDTA, 4 mM EGTA. CUT&RUN Stop buffer: Into 1 mL of 2X Stop buffer stock, add 5 μL of 10 mg/mL RNase A and 3.3 μL of 15 mg/mL. GlycoBlue Coprecipitant (Thermo Fisher AM9516)

### Antibodies

S7 Table lists all antibodies and conditions used in this study.

### ESC culture and mutagenesis

H9/WA09 (WiCell) hESCs were grown on Matrigel with reduced growth factors (Thermo Fisher Scientific, #35423) in mTeSR1 medium (STEMCELL Technologies, #85850) at 37°C and 5% $CO_2$. The 53BP1 knock-in cell lines (53BP1 S25A 34–3, 34–4, 79–1, 79–3 and S25D 14–3, 14–15, 14–19, 17) and ATM KO cell lines (ATM-KO2, 3, 14, and 43) were generated using CRISPR/Cas9 gene-editing technology. Genome editing reagents were designed and

validated in the Center for Advanced Genome Engineering at St. Jude Children's Research Hospital. Briefly, a chemically modified sgRNA (Synthego) was precomplexed with *SpCas9* protein (St. Jude Protein Production Core) and cotransfected with an ssODN donor template containing the desired modification into H9/WA09 cells via nucleofection (Amaxa P3 primary cell 4D nucleofector X kit L, Lonza) using the manufacturer's recommended protocol. Transfected cells were sorted (BD FACSAria Fusion) onto Matrigel and allowed to grown single-cell clones. Clones were identified via targeted mi-seq using a 2-step PCR library setup as previously described [37]. Samples were demultiplexed using the index sequences, fastq files were generated, and NGS analysis was performed using CRIS.py [38]. S8 Table lists genome editing reagents and associated primers.

## Neural progenitor cell generation and culture

ESCs were seeded onto AggreWell800 plates (STEMCELL Technologies, #34811) and fed with neural induction medium (STEMCELL Technologies, #05835) to form embryoid bodies. On day 5, embryoid bodies were replated onto Matrigel-treated 6-well plates in the same media. On day 17, cells were harvested as NPCs.

## Nuclear extract preparation and western blotting

ESCs and NPCs were incubated in Buffer A + PI + DTT for 5 min on ice. After centrifugation at 1,750*g* for 2 min at 4°C, the nuclei pellet was washed in Buffer A and subsequently incubated for approximately 25 min in Buffer D + PI + DTT at 4°C with rotation to obtain the nuclear fraction. Nuclear extracts were separated by SDS–PAGE and transferred onto a nitrocellulose membrane (Bio-Rad). Membranes were blocked with 3% bovine serum albumin (BSA) in HEPM, incubated in primary antibodies (HEPM containing 1% BSA and 0.1% Triton X-100) overnight at 4°C, washed in PBS-T, incubated in IRDye-conjugated secondary antibodies (LI-COR), and imaged on an Odyssey Fc imaging system (LI-COR). Signals were quantitated with the Image Studio software (version 1.0.14; LI-COR).

## Immunoprecipitation

Antibody was bound to protein A and protein G Dynabeads (Thermo Fisher 10002D and 10004D) for 2 h at room temperature. Nuclear extract was incubated with the Dynabeads-antibody complex for 5 h at 4°C, washed with PBST, and eluted with 0.1 M glycine (pH 2.3). Eluates were neutralized with 1/10 volume of 1.5 M Tris buffer (pH 8.8).

## Cortical organoid differentiation

Cortical organoids were generated based on previously published methods with minor modifications [39,40]. In brief, hESC lines were expanded and dissociated to single cells using Accutase, seeded onto low-attachment V-bottom 96-well plates (Costar, #7007) at a density of 9,000 cells per well to aggregate into embryoid bodies. The embryoid bodies formation medium (DMEM/F-12 with 20% KO serum replacement, 3% ESC-quality FBS, 2 mM GlutaMAX, 0.1 mM nonessential amino acids) was supplemented with dorsomorphin (2 μM), WNT inhibitor (IWR1, 3 μM), TGF-β inhibitor (SB431542, 5 μM), and Rho kinase inhibitor (Y-27623, 20 μM). Starting from day 4, embryoid bodies were fed with cortical differentiation medium (Glasgow-MEM, 20% KSR, 0.1 mM NEAA, 1 mM sodium pyruvate, 0.1 mM β-ME, and 1% anti-anti), supplemented with WNT inhibitor (IWR1, 3 μM), TGF-β inhibitor (SB431542, 5 μM), cyclopamine (2.5 μM) and Rho kinase inhibitor (Y-27623, 20 μM). On day 17, embryoid bodies were embedded in Matrigel droplets and transferred onto low-attachment 6-wells

and cultured in suspension using DMEM/F-12 supplemented with 1% N2 supplement, 1% lipid concentrate, 2% B27 supplement without vitamin A, and 1% anti-anti under 40% $O_2$/5% $CO_2$ conditions on shaker. Starting from day 30, medium was changed to 50% DMED/F-12, 50% neurobasal media, 0.5% N2 supplement, 1% GlutaMax, 0.05 mM NEAA, 0.025% human insulin, 0.1 mM β-ME, and 1% anti-anti, supplemented with 2% B27.

## Immunofluorescence

Cells and cryosectioned organoids were blocked with IF blocking solution for 2 h at room temperature and primary antibodies (diluted in blocking buffer) added and incubated O/N at 4°C. After 3 washes in PBS-T, fluorescent dye-conjugated secondary antibodies (1:500, Alexa Fluor-CONJUGATED antibodies, Thermo Fisher Scientific) were added and incubated for 3 h at room temperature. Secondary was washed with PBS-T 3 times, and samples were washed and coverslips mounted with Prolong Glass Mounting Reagent (Thermo Fisher Scientific), which contains DAPI. Images were acquired with Zeiss LSM780.

## Organoid feature characterization by image analysis

At days 35 and 55, bright-field images of organoids were captured with Axiocam 208 (Zeiss). Area of organoids, area of ventricular zone–like regions, and marker-positive cells were quantified by using the software FIJI: Signals-positive cells were identified based on signal and width thresholds. For ventricular zone–like region quantification, inner and outer edges of the regions in the image were manually traced, based on CTIP2-positive cells encircling the outer edges. FIJI was used to quantify area, perimeter, major and minor axes of the inner and outer traces. Mean perimeter and the difference between the major axes of the inner and outer traces were used to estimate the thickness of the structure. Mean Perimeter = (outer perimeter + inner perimeter) / 2. MajorAxisDiff = (outer major axis − inner major axis) / 2. MinorAxisDiff = (outer minor axis − inner minor axis) / 2. To quantify ZO-1-positive ventricular surfaces, ZO-1 signals were normalized by the Integral Image Filters plugin, and surface areas were manually traced for quantification. The VZ/SVZ structure was considered organized if PAX6-positive nuclei were densely packed with radial organization around ZO-1-positive ventricular surfaces. Ilastik [41] was used to quantify nuclear areas positive for different markers, using segmentation via a machine learning-based package and area quantification of segmented areas. Marker ratios were then calculated based on quantified areas.

## Quantification of cell populations by fluorescence-activated cell sorting (FACS)

Nine to 12 organoids of each line were dissociated using the papain dissociation system (Worthington LK003153). Dissociated cells were fixed in 4% formaldehyde solution at 4°C overnight and washed once in 1X PBS. Then, cells were permeabilized in 1X PBST for 2 h at room temperature on an orbital shaker. Cells were blocked in IF blocking buffer (1/3 Blocker Casein (Thermo Fisher 37528), 2/3 HEPM with 0.05% Triton X-100) for 2 h at room temperature on a shaker. Primary antibodies in IF blocking buffer were mixed with cells at 4°C overnight followed by washing twice with 1X PBST. Secondary antibodies in IF blocking buffer were mixed with cells for 2 h at room temperature on a shaker. After washing cells once, a conjugated antibody was added and incubated for 2 h at room temperature on a shaker. Cells were washed one last time before resuspended in 1X PBS for FACS. FACSymphony A1 sorter was used for analysis. All the centrifugation steps were done at $500 \times g$ for 4 min at room temperature. All washes were performed by incubating the cells with 1X PBS (after fixation) or PBST (after antibody staining) for 5 min at room temperature on a shaker. Primary antibodies used

are Ki67 (Cell Signaling 9129), PAX6 (DSHB supernatant 1mL), CTIP2 (Abcam 18465), and cleaved Caspase3-AF405-conjugated (R&D Systems IC835V).

## RNA-seq

Total RNA was extracted with TRIzol reagent (Invitrogen, #15596026) and Direct-zol RNA Microprep (Zymo Research, # R2062) by following manufacturer's instructions. DNA digestion with DNase I was performed during RNA extraction. Paired-end 100-cycle sequencing was performed on NovaSeq6000 sequencer by following the manufacturer's instructions (Illumina). Raw reads were first trimmed using TrimGalore (version 0.6.3) available at: https://www.bioinformatics.babraham.ac.uk/projects/trim_galore/, with parameters '—paired—retain_unpaired'. Filtered reads were then mapped to the *Homo sapiens* reference genome (GRCh38 + Gencode-v31) using STAR (version 2.7.9a) [42]. Gene-level read quantification was done using RSEM (version 1.3.1) [43]. To identify the differentially expressed genes between control and experimental samples, the variation in the library size between samples was first normalized by trimmed mean of $M$ values (TMM) and genes with CPM < 1 in all samples were eliminated. Then, the normalized data were applied to linear modeling with the voom from the limma R package [44]. GSEA was performed against using the MSigDB database (version 7.1), and differentially expressed genes were ranked based on $\log_2$(FC) [45,46].

## Protein extraction, digestion, and Tandem-Mass-Tag (TMT) labeling

Organoids were harvested on day 35, and the Matrigel droplets were eliminated by multiple ice-cold PBS washes. The organoid pellet was extracted in the lysis buffer (50 mM HEPES (pH 8.5), 8 M urea, and 0.5% sodium deoxycholate, 100 µl buffer per 10 mg tissue) with 1x PhosSTOP phosphatase inhibitor cocktail (Sigma-Aldrich). Protein concentration was estimated by a Coomassie stained short gel with BSA as a standard. About 600 µg each of protein samples was digested with LysC (Wako) at an enzyme-to-substrate ratio of 1:100 (w/w) for 2 h at room temperature in the presence of 1 mM DTT. The samples were then diluted to a final 2 M urea concentration with 50 mM HEPES (pH 8.5) and digested with Trypsin (Promega) at an enzyme-to-substrate ratio of 1:50 (w/w) for 3 h. The peptides were reduced by adding 1 mM DTT for 30 min at room temperature followed by alkylation with 10 mM iodoacetamide for 30 min in the dark at room temperature. The unreacted iodoacetamide was quenched with 30 mM DTT for 30 min. Finally, the digestion was terminated and acidified by adding trifluoroacetic acid to 1%, peptides desalted using Sep-Pak C18 cartridge (Waters), and dried by speed vac. The purified peptides were resuspended in 50 mM HEPES (pH 8.5) and labeled with 16-plex Tandem Mass Tag (TMTpro) reagents (Thermo Scientific) following the manufacturer's recommendation. The TMT labeled samples were mixed equally, desalted using Sep-Pak C18 cartridge (Waters), and dried by speed vac.

## Offline fractionation and two-dimensional liquid chromatography-tandem mass spectrometry (LC/LC-MS/MS)

The dried TMT mix was resuspended and fractionated on an offline HPLC (Agilent 1220) using basic pH reverse phase liquid chromatography (pH 8.0, XBridge C18 column, 4.6 mm × 25 cm, 3.5 µm particle size, Waters). A total of 160 one-minute fractions were collected and concatenated to 80 fractions. For whole proteome analysis, 10% of these 80 fractions was used. The remaining 90% of the 80 fractions were concatenated to 20 fractions for phophopeptide enrichment. Phosphopeptide enrichment was performed according to a previously published protocol [47]. The phosphopeptide enrichment eluents and the total proteome fractions were dried and resuspended in 5% formic acid and analyzed by acidic pH reverse phase LC-MS/MS analysis. The peptide samples were loaded on a nanoscale capillary reverse phase

C18 column (New objective, 75 μm ID × approximately 15 cm, 1.9 μm C18 resin from Dr. Maisch GmbH) by a HPLC system (Thermo Ultimate 3000) and eluted by either a 125-min gradient (phosphofractions) or 110-min gradient for total proteome fractions. The eluted peptides were ionized by electrospray ionization and detected by an inline Orbitrap Fusion mass spectrometer (Thermo Scientific). For total proteome fractions, the mass spectrometer is operated in data-dependent mode with a survey scan in Orbitrap (60,000 resolution, $2 \times 10^5$ AGC target and 50 ms maximal ion time) and MS/MS high-resolution scans (60,000 resolution, $1 \times 10^5$ AGC target, 150 ms maximal ion time, 36.5 HCD normalized collision energy, 1 $m/z$ isolation window, and 15-s dynamic exclusion). For phosphoproteome fractions, the mass spectrometer is operated in data-dependent mode with a survey scan in Orbitrap (60,000 resolution, $3 \times 10^5$ AGC target and 50 ms maximal ion time) and MS/MS high-resolution scans (60,000 resolution, $1 \times 10^5$ AGC target, 150 ms maximal ion time, 36.5 HCD normalized collision energy, 1 $m/z$ isolation window, and 10-s dynamic exclusion).

### Identification of proteins and phosphopeptides

The MS/MS raw data were processed by a tag-based hybrid search engine JUMP [48]. The data were searched against the UniProt human database (168,305 protein entries; downloaded in April 2020) concatenated with a reversed decoy database for evaluating FDR. Searches were performed using a 15-ppm mass tolerance for fragment ions, fully tryptic restriction with 2 maximal missed cleavages, 3 maximal modification sites, and the assignment of $b$ and $y$ ions. TMT tags on Lysine residues and N-termini (+304.2071453 Da) were used for static modifications and Met oxidation (+15.99492 Da) was considered as a dynamic modification. Phosphorylation (+79.96633 Da) was considered as a dynamic modification for STY residues. Putative peptide spectral matches (PSMs) were filtered by mass accuracy and then grouped by precursor ion charge state and filtered by JUMP-based matching scores (Jscore and ΔJn) to reduce FDR below 1% for proteins during the whole proteome analysis or 1% for phosphopeptides during the phosphoproteome analysis. Phosphosites were further evaluated by JUMPl program using the concept of the phosphoRS algorithm [49] to calculate phosphosite localization scores (Lscore, 0% to 100%) for each PSM.

### Quantification of proteins and phosphopeptides

TMT reporter ion intensities of each PSM were extracted and corrected based on isotopic distribution of each labeling reagent. Those PSMs with very low intensities (e.g., minimum intensity of 1,000 and median intensity of 5,000) were excluded for quantification. Sample loading bias was mitigated by normalization with the trimmed median intensity of all PSMs. Protein or phosphopeptide relative intensities were calculated by dividing the intensity of each channel by the mean intensity. Protein or phosphopeptide absolute intensities were computed by multiplying the relative intensities by the grand-mean of 3 most highly abundant PSMs.

### Differential expression analysis of proteins and phosphopeptides

Differentially expressed proteins between the 2 strains and 2 different doses were identified by the limma R package [50]. The Benjamini–Hochberg method was used to control multiple-testing correction, and proteins with an adjusted $p$-value of <0.05 and log2 fold change of > 1.5 were defined as differentially expressed.

### Pathway enrichment analysis for proteomics data

Pathway enrichment analysis was carried out to infer functional groups of proteins that were enriched in a given dataset. The 4 common pathway databases were used, including Gene

Ontology (GO), KEGG, Hallmark, and Reactome. The analysis was performed using Fisher's exact test with the Benjamini–Hochberg correction for multiple testing. A cutoff of adjusted *p*-value < 0.2 was used to identify significantly enriched pathways.

### Estimation of kinase activity

Kinase activity was inferred based on known substrates in the PhosphoSitePlus database [51] using the IKAP algorithm [24]. The phosphoproteome data were normalized against the whole proteome. We performed 100 times of calculations to overcome the potential problem of local optimization.

### Chromatin immunoprecipitation

Cells were harvested in PBS. Cytoplasmic fractions were extracted using buffer A with 1× protease inhibitors and 1 mM DTT. Nuclear pellets were cross-linked by 1.1% formaldehyde in buffer B with 1× protease inhibitors and 1 mM DTT; washed; and lysed in lysis buffer 3 with 1× protease inhibitors, 1 mM DTT, and 1 mM PMSF. The fixed and lysed nuclear extract was sonicated with Bioruptor Pico (Diagenode) 10 times for 15 s each, with 45-s intervals. Chromatin was added to Dynabeads (Life Technologies) prebound with 4 μg of antibodies for overnight incubation. After incubation, beads were washed and immunoprecipitates were eluted. DNA from eluates was recovered by the GeneJET FFPE DNA purification kit (Thermo Fisher Scientific, #K0882). DNA libraries were generated using the NEBNext Ultra DNA Library Prep kit (NEB, #E7370S) and sequenced at the St. Jude Hartwell Center.

### CUT&RUN

Approximately $5 \times 10^5$ live cells were mixed with $5 \times 10^4$ *Drosophila* S2 cells per reaction. For CUT&RUN, we followed EpiCypher CUTANA protocol. In brief, we first isolated nuclei by incubating cells on ice for 5 min in Buffer A with protease inhibitor and 0.1% Triton X-100. After centrifugation at $1,750 \times g$ for 2 min at 4˚C, nuclei were resuspended in Wash buffer. Bio-Mag Plus Concanavalin-A (Con A) coated beads (Bangs Laboratories BP531) activated in Binding buffer were then added to the nuclei and rotated for 10 min at room temperature. About 1 μg primary antibody with 0.25 μg Spike-in antibody (Active Motif 61686) diluted in Antibody buffer was added to the bead-nuclei mixture and incubated for 2 h at room temperature. Beads were washed twice with Digitonin buffer and incubated with pAG-MNase for 10 min at room temperature. Beads were then washed twice with Digitonin buffer, incubated with 2 mM $CaCl_2$ for 2 h at 4˚C, and quenched by adding Stop buffer. DNA was released from the beads by incubating them for 10 min at 37˚C and purified by CUTANA DNA purification kit (EpiCypher SKU:14–0050). Libraries were constructed using xGen ssDNA and Low-Input DNA Prep by following the manufacturer's instructions (IDT 10009817) and sequenced at the St. Jude Hartwell Center.

### Analysis of chromatin immunoprecipitation-sequencing and CUT&RUN

Approximately 50 bp single-end reads were obtained and aligned to human genome hg38 by BWA (version 0.7.170.7.12, default parameter). Duplicated reads were marked by the bamsormadup from the biobambam tool (version 2.0.87) available at https://www.sanger.ac.uk/tool/biobambam/. Uniquely mapped reads were kept by samtools (parameter "-q 1 -F 1804," version 1.14). Fragments <2,000 bp were kept for peak calling, and bigwig files were generated for visualization. SICER [27] and macs2 [28] were both used for peak calling to identify both the narrow and broad peak correctly. With SICER, we assigned peaks that were at the top 1

percentile as the high-confidence peaks and the top 5 percentile as the low-confidence peaks. Two sets of peaks were generated: Strong peaks called with parameter "FDR < 0.05" by at least 1 method (macs2 or SICER) and weak peaks called with parameter "FDR < 0.5" by at least 1 method (macs2 or SICER). Peaks were considered reproducible if they were supported by 1 strong peaks and at least 1 weak peak in other replicates. For downstream analyses, heatmaps were generated by deepTools [52], and gene ontology was performed with Enrichr [53,54] and GSEA, in addition to custom R scripts. For differential peak analysis, peaks from 2 replicates were merged and counted for number of overlapping extended reads for each sample (bedtools v2.24.0) [55]. Then, we detected the differential peaks by the empirical Bayes method (eBayes function from the limma R package) [44]. For downstream analyses, heatmaps were generated by deepTools (v3.5.0) [56]. Peaks were annotated based on Gencode following this priority: "Promoter.Up": if they fall within TSS– 2 kb, "Promoter.Down": if they fall within TSS– 2 kb, "Exonic" or "intronic": if they fall within an exon or intron of any isoform, "TES peaks": if they fall within TES ± 2 kb, "distal5" or "distal3" if they are with 50 kb upstream of TSS or 50 kb downstream of TES, respectively, and they are classified as "intergenic" if they do not fit in any of the previous categories.

## Supporting information

**S1 Data. Numerical data used to generate summary data in this study.**
(XLSX)

**S1 Raw Images. Uncropped western blot images in this study.**
(PDF)

**S1 Fig. Characterization of 53BP1-pS25 and NPCs and genome editing of hESCs.** (**A**) Schematic diagram of neural differentiation of hESCs: neural induction, differentiation, and maturation media to form EBs, rosettes, NPCs, and neurons. (**B**) Principal component analysis of WT ESCs, NPCs, day 10 (D10) cortical organoids, and D17 cortical organoids. GSEA terms that are highly enriched in significantly (**C**) down-regulated and (**D**) up-regulated genes in WT NPCs compared to ESCs. % Match, % of genes in the enriched term that overlap the differentially expressed genes or proteins. (**E**) Immunofluorescence of NPC markers PAX6 and NESTIN. Bar, 50 μm. (**F**) Quantification of 53BP1-pS25-positive hESCs or hNPCs. Data are presented as the mean ± SEM, with $p < 0.0001$. (**G**) WB analysis of control cells and 53BP1-KO clones 415, 416, and 209, which are clones KO1, KO2, and KO3 in Yang and colleagues' study [6]. (H) WB analysis of control and 53BP1-S25A hNPCs. The S25A mutation prohibits phosphorylation. (**I**) WB analysis of hESCs and hNPCs and quantification. (**J**) Schematic diagram of genome editing in hESCs. Guide RNA 6 were complexed with Cas9 proteins and used along single-stranded nucleotide donors to transfect hESCs. Individual clones from transfection were cultured, sequenced by mi-seq across the targeted *53BP1* locus, and established as >99% pure clonal lines. Diagram was generated using open-sourced images available at biorender.com. Underlying numerical values for figures are found in S1 Data. EB, embyoid body; ESC, embryonic stem cell; GSEA, gene set enrichment analysis; hESC, human embryonic stem cell; hNPC, human neural progenitor cell; KO, knockout; NES, normalized enrichment score; NPC, neural progenitor cell; WB, western blot; WT, wild type; 53BP1-pS25, 53BP1 phosphorylated at serine 25.
(PDF)

**S2 Fig. Generation and analyses of ATM-KO hESCs and cortical organoids.** (**A**) Alignment of WT and *ATM*-KO mutation sequences on 2 alleles (al) in the *ATM* locus. Red indicates the gRNA sequence. (**B**) WB analysis of WT and 4 *ATM*-KO hNPCs. (**C**) Principal component

analysis showed the intermixing and similar RNA-seq profiles from hESCs of 7 WT, 4 53BP1-S25A, 4 53BP1-S25D, 4 ATM-KO, and 4 53BP1-KO lines. (**D**) Immunofluorescence showed similar expression of OCT4 and SSEA4 proteins in control and *ATM*-KO hESCs. Bar, 100 μm. (**E**) WB analysis of WT and 2 *ATM*-KO hNPCs. Quantification suggests reduction of γH2AX in *ATM*-KO hNPCs. Welch's *t* test was used to perform pairwise comparisons of WT and *ATM*-KO. Underlying numerical values for figures are found in S1 Data. ATM, ataxia telangiectasia mutated; hESC, human embryonic stem cell; hNPC, human neural progenitor cell; KO, knockout; WB, western blot; WT, wild type.
(PDF)

**S3 Fig. Analysis of γH2AX and CC3 in cortical organoids.** (**A**) Immunofluorescence showed D35 *ATM*-KO and WT cortical organoids had similar γH2AX foci. Bar, 100 μm. FACS analysis of CC3 in (**B**) D21 and (**C**) D28 cortical organoids. Two biological replicates were done, and each data point was based on 3 technical replicate analyses of 10–12 cortical organoids. (**D**, **E**) Immunofluorescence and quantification of CC3 in D28 cortical organoids. Bar, 100 μm. Graphs are presented in ratios (out of 1), with **, $p < 0.01$; ****, $p < 0.0001$; ns, not significant by two-way ANOVA test. Underlying numerical values for figures are found in S1 Data. ATM, ataxia telangiectasia mutated; CC3, cleaved-caspase 3; KO, knockout; WT, wild type.
(PDF)

**S4 Fig. Analysis of cell proliferation in cortical organoids.** FACS analysis of PAX6 and KI67 in (**A**, **B**) D28 and (**C**) D35 cortical organoids. Each data point was based on the 3 technical replicate analyses of 10 cortical organoids. (**D**) Quantification of KI67/PAX6 ratios in immunofluorescence of D35 cortical organoids. Each data point represents quantification of cells in 1 cortical organoid. (**E**, **F**) Immunofluorescence and quantification of H3-pS10 (PH3) in D28 cortical organoids. Bar, 100 μm. (**F-H**) Immunofluorescence and quantification of PH3 and KI67 in D35 cortical organoids. Bar, 100 μm. ***, $p < 0.001$; ns, not significant by two-way ANOVA test. Underlying numerical values for figures are found in S1 Data.
(PDF)

**S5 Fig. Immunofluorescence analyses of cortical organoids and NPCs.** Immunofluorescence of (**A**) NEUN and (**E**) ZO-1 and PAX6 in D37 cortical organoids. Bar, 100 μm. (**B**) Immunofluorescence of ZO-1 in D28 cortical organoids. Bar, 100 μm. Quantification of the (**C**) number and (**D**) surface area of ZO-1-positive ventricles in D28 cortical organoids. *, $p < 0.05$; ***, $p < 0.001$; ns, not significant by two-way ANOVA test. (**F**) Bright-field images of cortical organoids formed by *ATM*-KO2, 3, 14, 43, and WT control at day 55 of differentiation. Bar, 1.5 mm. (**G**) The size of cortical organoids was compared between groups by one-way ANOVA with Dunnett's multiple comparisons test, with ns, not significant and ***, $p < 0.001$. $n = 13$ organoids/group. Underlying numerical values for figures are found in S1 Data. ATM, ataxia telangiectasia mutated; NPC, neural progenitor cell; WT, wild type.
(PDF)

**S6 Fig. Characterization of NPCs and D35 cortical organoids.** (**A**) Immunofluorescence of PAX6 and NES in NPCs. Bar, 50 μm. (**B**) Principal component analysis of proteomics data of D35 WT and *ATM*-KO cortical organoids. GSEA terms that are highly enriched in significantly (**C**) higher and (**D**) lower total proteins in D35 *ATM*-KO versus WT cortical organoids. (**E**) GSEA terms that are highly enriched in significantly higher phosphoproteins, which were normalized to total proteomics, in D35 *ATM*-KO versus WT cortical organoids. Underlying numerical values for figures are found in S1 Data. ATM, ataxia telangiectasia mutated; GSEA, gene set enrichment analysis; KO, knockout; NES, normalized enrichment score; NPC, neural

progenitor cell; WT, wild type.
(PDF)

**S7 Fig. Kinase activities in cortical organoids and characterization of the 53BP1-S25A and 53BP1-S25D hESCs.** Heatmaps showing relative phosphorylation levels of (**A**) 7 MAPK9 substrates that are significantly lower and (**B**) 7 CDK5 substrates that are significantly higher in D35 *ATM*-KO versus WT cortical organoids. (**C**) Heatmaps showing activity of selected protein kinases between ATM-KO3, ATM-KO4, and WT cell lines. (**D**) Alignment of WT and 53BP1-S25A and S25D mutation sequences on 2 alleles (al). Red indicates the gRNA sequence. Underline indicates codon encoding the WT serine 25, mutant alanine, or mutant aspartic acid. (**E**) WB analysis of control and 53BP1-S25D hNPCs, which have comparable levels of 53BP1 protein. (**F**) Transcripts per million values of 10 pluripotency genes were used for comparison to show that control, 53BP1-S25A, and 53BP1-S25D hESCs did not differ in pluripotency. Underlying numerical values for figures are found in S1 Data. ATM, ataxia telangiectasia mutated; hESC, human embryonic stem cell; hNPC, human neural progenitor cell; KO, knockout; WB, western blot; WT, wild type; 53BP1, p53 binding protein 1.
(PDF)

**S8 Fig. Characterization of the 53BP1-S25A and 53BP1-S25D hESCs and cortical organoids.** (**A**) Immunofluorescence showed similar expression of OCT4 and SSEA4 proteins in WT, 53BP1-S25A, and 53BP1-S25D hESCs. Bar, 100 μm. Immunofluorescence of (**B**) KI67 and (**D**) PH3 in cryosections of cortical organoids at day 35 of differentiation. Bar, 100 μm. (**C**) Quantification of KI67-positive cells in D35 cortical organoids. Data points represent single organoids. The mean ± SEM values were compared by one-way ANOVA with Dunnett's multiple comparisons test to yield ****, ***, and ** indicating $p < 0.0001$, 0.001, and 0.01, respectively. $n = 3$ organoids/group. Underlying numerical values for figures are found in S1 Data.
(PDF)

**S9 Fig. Analysis of γH2AX and CC3 in cortical organoids.** Immunofluorescence of (**A**) γH2AX in D35 cortical organoids and (**D**) CC3 in D28 cortical organoids. Bar, 100 μm. CC3 quantification by FACS of (**B**) D21 and (**C**) D28 cortical organoids. For each datapoint, 10–12 organoids from each line were analyzed via 3 technical replicates, and data from 4 mutant lines were consolidated to achieve rigorous comparisons. **, $p < 0.01$ and ns, not significant by two-way ANOVA test. (**E**) CC3 quantification of immunofluorescence images of D28 cortical organoids. For each line, 4–6 images and >10,000 cells were analyzed. *, $p < 0.05$; **, $p < 0.01$; ns, not significant by two-way ANOVA test. Graphs in (**B**, **C**, **E**) are presented in ratios (out of 1). Underlying numerical values for figures are found in S1 Data.
(PDF)

**S10 Fig. 53BP1-pS25 promotes ventricle formation in cortical organoids.** Quantification of the (**A**) number and (**B**) surface area of ZO-1-positive ventricles. (**C**) Immunofluorescence of ZO-1 in D28 cortical organoids. Bar, 100 μm. *, $p < 0.05$; **, $p < 0.01$; ***, $p < 0.001$; ns, not significant by two-way ANOVA test. Underlying numerical values for figures are found in S1 Data.
(PDF)

**S11 Fig. Characterization of 53BP1-S25A and 53BP1-S25D cortical organoids.** (**A**) Brightfield images of cortical organoids formed by cell lines 53BP1-S25A 34–3, 34–4, 79–1, 79–3 and S25D 14–3, 14–15, 14–19, 17, and 2 WT control at day 55 of differentiation. Bar, 1.5 mm. Blue transparent structures around organoids are Matrigel embedment. (**B**) At day 55 of

differentiation, the size of cortical organoids was compared between groups. (**C**) The growth (comparing organoids at days 35 and 55) of cortical organoids were compared between groups. Data points represent single organoids. The mean ± SEM values were compared by one-way ANOVA with Dunnett's multiple comparisons test to yield **** and ** indicating $p < 0.0001$ and 0.01, respectively. $n = 15$–36 organoids/group. (**D**) Two genes overlapped between up-regulated genes in 53BP1-S25A versus WT and down-regulated genes in 53BP1-S25D versus WT cortical organoids. No gene overlapped between down-regulated genes in 53BP1-S25A versus WT and up-regulated genes in 53BP1-S25D versus WT cortical organoids. (**E**) Down-regulated GSEA terms between 53BP1-S25A versus WT and 53BP1-S25D versus WT were not highly overlapped. Ten GSEA terms were specific to 53BP1-S25D versus WT. Underlying numerical values for figures are found in S1 Data. (PDF)

**S12 Fig. Characterization of NPCs and comparative analyses of RNA-seq data.** (**A**) Immunofluorescence of NPC markers PAX6 and NESTIN. Bar, 50 μm. GSEA identified top enrichment of differentially expressed genes in (**B, D**) 53BP1-S25A or (**C, E**) S25D versus WT NPCs. % Match, % of genes in the enriched term that overlap the differentially expressed genes or proteins. Venn diagrams depict overlaps between down-regulated genes in *ATM*-KO with 53BP1- (**F**) S25A or (**G**) S25D cortical organoids. Underlying numerical values for figures are found in S1 Data. ATM, ataxia telangiectasia mutated; GSEA, gene set enrichment analysis; KO, knockout; NES, normalized enrichment score; NPC, neural progenitor cell; WT, wild type. (PDF)

**S13 Fig. Comparisons of RNA-seq data and 53BP1 ChIP-seq analyses.** (**A**) GSEA graphs showed that up-regulated genes in 53BP1-S25A or S25D vs. WT had significant enrichment in down-regulated genes of ATM-KO vs. WT cortical organoids. *P* values were calculated by the hypergeometric test, assuming normal data distribution. (**B**) Concordantly differential expression of genes in 53BP1-S25D vs. WT were enriched in those in 53BP1-S25A vs. WT. (**C**) Concordantly differential expression of genes in 53BP1-S25A vs. WT were enriched in those in 53BP1-S25D vs. WT. For (**A-C**), *P* values were calculated by the hypergeometric test, assuming normal data distribution. (**D**) Proportions of 53BP1 binding to genomic features. 53BP1 ChIP-seq tracks at loci of representative (**E**) up-regulated and (**F**) down-regulated genes in 53BP1-S25A and S25D versus WT D35 cortical organoids. (**G**) S25A and S25D down-regulate 53BP1 targets that are enriched in IRE1-mediated unfolded protein response, regulation of cellular response to stress, iron import into cells, and regulation of apoptosis. Underlying numerical values for figures are found in S1 Data. ATM, ataxia telangiectasia mutated; GSEA, gene set enrichment analysis; KO, knockout; WT, wild type. (PDF)

**S14 Fig. 53BP1 ChIP-seq and 53BP1-pS25 CUT&RUN.** (**A**) MA plot displays 53BP1 ChIP-seq signals at genomic sites that are significantly different in *ATM*-KO vs. WT NPCs. Proportions of genomic features and gene ontology of genes with (**B**) higher or (**C**) lower 53BP1 binding in *ATM*-KO vs. WT NPCs. (**D**) Proportions of 53BP1-pS25 binding to genomic features. (**E**) GSEA identified top enrichment of genes occupied by 53BP1-pS25 in WT NPCs. % Match, % of genes in the enriched term that overlap the differentially expressed genes or proteins. Underlying numerical values for figures are found in S1 Data. ATM, ataxia telangiectasia mutated; GSEA, gene set enrichment analysis; KO, knockout; NES, normalized enrichment score; NPC, neural progenitor cell; WT, wild type. (PDF)

**S15 Fig. Differential 53BP1 ChIP-seq in 53BP1-WT, S25A, and S25D NPCs.** (**A**) Principal component analysis of top 3,000 most variable peaks in 53BP1 ChIP-seq of 53BP1-WT, S25A, and S25D NPCs. Two independent cell lines for each group were used for ChIP-seq. Proportions of genomic features in regions with significantly different 53BP1 ChIP-seq in (**B**) 53BP1-S25A vs. WT and (**C**) 53BP1-S25D vs. WT, using the criterion of FC>2 and $p < 0.05$. (**D**) Heatmaps aligning peaks with significantly different 53BP1 ChIP-seq in 53BP1-S25A vs. S25D. Control regions are those, after voom normalization, showed the least changes and served as semi-independent validation of differential ChIP-seq analysis. Bubble graphs present top enriched categories of genes that had significantly lower 53BP1 ChIP-seq in (**E**) 53BP1-S25A vs. WT and (**F**) 53BP1-S25D vs. WT. Underlying numerical values for figures are found in S1 Data. FC, fold-change; NPC, neural progenitor cell; WT, wild type.
(PDF)

**S16 Fig. Analysis of ATM activities during the inhibition of TGFβ, WNT, and HH signaling.** (**A**) Schematic diagram of neural specification of hESCs with HH (SB421542), TGFβ (dorsomorphin), and WNT (IWR1e and cyclopamine) signaling inhibitors. Nuclear extract was harvested on day 4 and day 10. (**B**) WB analysis of day 4 samples. (**C**) Quantification of day 4 WB. Data are presented as the mean ± SEM, and Student $t$ test was performed for pairwise comparisons. n.s., *, and ** indicate not significant, $p < 0.05$, and $p < 0.01$, respectively. (**D**) WB analysis of day 10 samples. (**E**) Quantification of day 10 WB. Data are presented as the mean ± SEM, and Student $t$ test was performed for pairwise comparisons. n.s., *, and ** indicate not significant, $p < 0.05$, and $p < 0.01$, respectively. Underlying numerical values for figures are found in S1 Data. ATM, ataxia telangiectasia mutated; hESC, human embryonic stem cell; WB, western blot.
(PDF)

**S17 Fig. *RNF168*-KO alters key genetic programs and 53BP1-pS25 binding to chromatin.** (**A**) Alignment of WT and *RNF168*-KO mutation sequences in the *RNF168* locus. Red indicates the gRNA sequences. (**B**) WB analysis of WT and *RNF168*-KO hESCs. (**C**) RT-qPCR analysis showing that pluripotent genes in *RNF168*-KO were expressed higher or the same as those in WT. *RNF168*-KO did not reduce pluripotent gene expression. *, $p < 0.05$; ns, not significant by two-way ANOVA text. (**D**) Immunofluorescence showed similar expression of OCT4 and SSEA4 proteins in WT and *RNF168*-KO hESCs. Bar, 100 μm. (**E**) Immunofluorescence of showed similar expression of PAX6 and NES in NPCs. WT and *RNF168*-KO NPCs. Bar, 50 μm. Functional terms that are highly enriched in (**F**) up-regulated and (**G**) down-regulated genes in *RNF168*-KO D35 cortical organoids. % Match, % of genes in the enriched term that overlap the differentially expressed genes or proteins. (**H**) Heatmaps aligning peaks with 53BP1-pS25 CUT&RUN signals that were gained, the same, or lost in *RNF168*-KO vs. WT NPCs, using the criterion of FC>2 and $p < 0.05$. n = numbers of peaks. Regions with the same signals, are $n = 899$, which showed the least changes after voom normalization and served as semi-independent validation of differential ChIP-seq analysis. (**I**) Functional terms of 53BP1-pS25-bound genes in WT NPCs. % Match, % of genes in the enriched term that overlap the differentially bound genes. (**J**) Number of differentially expressed genes identified by comparison of *RNF168*-KO vs. WT NPCs at $p < 0.05$. Of these genes, we list the numbers of 53BP1-pS25-bound targets and targets with higher or lower 53BP1-pS25 CUT&RUN signals in *RNF168*-KO NPCs. Underlying numerical values for figures are found in S1 Data. FC, fold-change; hESC, human embryonic stem cell; KO, knockout; NES, normalized enrichment score; NPC, neural progenitor cell; RT-qPCR, quantitative reverse transcription PCR; WB, western blot; WT, wild type; 53BP1-pS25, 53BP1 phosphorylated at serine 25.
(PDF)

**S1 Table. All cell lines generated for this study were treated with trypsin and Wright's stain and then analyzed by the Cytogenetic Shared Resource at St. Jude.** Typically normal karyotypes and 3 abnormalities are shown.
(PDF)

**S2 Table. The expression of forebrain, midbrain, and hindbrain markers in D35 WT and *ATM*-KO cortical organoids.** Data suggest that D35 *ATM*-KO cortical organoids specified to the forebrain lineage.
(PDF)

**S3 Table. List of phosphoproteins, normalized to total protein levels, that were significantly lower in D35 *ATM*-KO versus WT cortical organoids.**
(XLSX)

**S4 Table. Normalized (to total proteome) levels of phosphopeptide substrates of MAPK9 and CDK5 in D35 WT and *ATM*-KO cortical organoids.**
(XLSX)

**S5 Table. Two-sample *t* test examines the sizes of cortical organoids that change between day 35 and day 55 of differentiation.** Data from WT and 53BP1 mutants are compared pairwise by using the two-sample *t* test. The sizes of organoids are significantly different between each comparison pair (all $p < 0.05$).
(PDF)

**S6 Table. The changes in organoid size at days 35 and 55 of differentiation were compared to yield S3C Fig.** This table lists the calculation for different combinations of data and the descriptive statistics.
(PDF)

**S7 Table. Primary antibodies used in this study.**
(PDF)

**S8 Table. gRNA sequences used in this study.**
(PDF)

## Acknowledgments

The authors thank A. Andersen and I. Chen for discussions and editing the manuscript; A. N. Kettenbach for advice; J. Houston and K. Lowe for FACS; P. Sinojia and E. Rivera-Peraza for preliminary experiments and data analyses. Sequencing was performed at the Harwell Center for Biotechnology, images were acquired at the Cell & Tissue Imaging Center, and karyotyping was analyzed by J. Wilbourne and V. Valentine at the Cytogenetics Core; all are supported by SJCRH and NCI P30 (CA021765).

## Author Contributions

**Conceptualization:** Jamy C. Peng.

**Data curation:** Bitna Lim, Yurika Matsui, Nina Connolly, Kanisha Kavdia.

**Formal analysis:** Bitna Lim, Yurika Matsui, Seunghyun Jung, Mohamed Nadhir Djekidel, Wenjie Qi, Zuo-Fei Yuan, Xusheng Wang, Xiaoyang Yang, Nina Connolly, Abbas Shirinifard Pilehroud, Fang Wang, Beisi Xu, Jamy C. Peng.

**Funding acquisition:** Jamy C. Peng.

**Investigation:** Bitna Lim, Jamy C. Peng.

**Methodology:** Yurika Matsui, Shondra M. Pruett-Miller, Jamy C. Peng.

**Project administration:** Jamy C. Peng.

**Resources:** Jamy C. Peng.

**Supervision:** Haitao Pan, Vishwajeeth Pagala, Yiping Fan, Junmin Peng, Beisi Xu, Jamy C. Peng.

**Visualization:** Bitna Lim, Yurika Matsui, Jamy C. Peng.

**Writing – original draft:** Jamy C. Peng.

**Writing – review & editing:** Yurika Matsui, Jamy C. Peng.

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
