## [Editor Report · Decision Letter 0]

4 Sep 2023

Dear Dr Peng, 

Thank you for submitting your manuscript entitled "Phosphorylation of 53BP1 by ATM controls neurodevelopmental programsin cortical brain organoids" for consideration as a Research Article by PLOS Biology.

Your manuscript has now been evaluated by the PLOS Biology editorial staff as well as by an academic editor with relevant expertise and I am writing to let you know that we would like to send your submission out for external peer review.

Once your full submission is complete, your paper will undergo a series of checks in preparation for peer review. After your manuscript has passed the checks it will be sent out for review. To provide the metadata for your submission, please Login to Editorial Manager (https://www.editorialmanager.com/pbiology) within two working days, i.e. by Sep 06 2023 11:59PM.

Kind regards,

Christian

Christian Schnell, Ph.D.

Senior Editor

PLOS Biology

cschnell@plos.org

---

## [Decision Letter · Decision Letter 1]

3 Nov 2023

Dear Dr Peng,

Thank you for your patience while your manuscript "Phosphorylation of 53BP1 by ATM controls neurodevelopmental programs in cortical brain organoids" was peer-reviewed at PLOS Biology. It has now been evaluated by the PLOS Biology editors, an Academic Editor with relevant expertise, and by several independent reviewers. In light of the reviews, which you will find at the end of this email, we would like to invite you to revise the work to thoroughly address the reviewers' reports.

As you will see below, the reviewers appreciate the importance of the topic examined here and comment that the study offers some potentially interesting insights, but they have also highlighted that additional experimental work is needed to bolster the conclusions of the study. After discussion with the Academic Editor, we think these requests will need to be thoroughly addressed before we can consider your manuscript further for publication at PLOS Biology. 

While we think it will be essential to expand the study, with new data and phenotypic analyses, we would not strictly require that you characterize an additional ES line as suggested by Reviewer 2. While we appreciate that adding these data would strengthen the study, we think that this request is beyond the scope of the current work. Additionally, we would not require the generation of new single cell RNA transcriptomics, but we do agree with the reviewers that more targeted analyses should be performed.

Given the extent of revision needed, we cannot make a decision about publication until we have seen the revised manuscript and your response to the reviewers' comments. Your revised manuscript is likely to be sent for further evaluation by all or a subset of the reviewers.

**IMPORTANT - SUBMITTING YOUR REVISION**

*Re-submission Checklist*

*Published Peer Review*

*PLOS Data Policy*

*Blot and Gel Data Policy*

Sincerely,

Lucas

Lucas Smith, Ph.D.

Senior Editor

PLOS Biology

lsmith@plos.org

REVIEWS:

Reviewer #1: This manuscript from Peng and colleagues investigate the role of ATM-dependent 53BP1 ser25 phosphorylation in neural development. Significantly, this work represents the follow up to prior work from the same group (Nature Neuroscience 2019), in which they reported a role for promoter bound 53BP1 - UTX complexes in upregulating neurodevelopmental genes during the differentiation of human embryonic stem cells into neurons or into cortical organoids.

Here the paper starts with the observation that 53BP1 expression doesn't change during differentiation from hESCs to neurons/cortical organoids, so they speculate it might instead be controlled by 53BP1 phosphorylation ATM, and then concentrate on the well-known Ser25-phosphosite in 53BP1 (a known ATM target) since this modification appears to be reduced in differentiated NPCs relative to hESCs in western blotted lysates. In correlation of such a notion, they find ATM-KO cells also show perturbed/modified transcriptional expression profiles during the differentiation experiments, they show ATM interacts with 53BP1 (by IP), and they show NPCs with 53BP1 Ser25Ala/Asp knockin mutations also have altered expression profiles, including alterations in neural gene expression profiles. Lastly, they correlate these changes to the altered residency of 53BP1 -S25 mutant proteins on specific gene promoters relative to wild type 53BP1 (which over-accumulates relative to WT), on which they speculate that ATM-dependent 53BP1 phosphorylation, guided by upstream DDR signalling incl ATM-dependent gH2AX induction at TSSs which recruits 53BP1, then negatively regulates 53BP1 occupancy on bivalent gene promoters, thereby modulating expression and contributing to neuronal differentiation.

The paper is quite descriptive and correlative in nature, with the bulk of data using transcriptomics and proteomics to identify changes in protein/gene expression between the different genetic backgrounds, GSEA/similar to attribute this to different gene classes that include (but are not limited to) genes involved in neural differentiation. In then showing that differences occur both in ATM and 53BP1 mutant cells, there may be some correlation/overlap, but it is difficult to interpret this as the stated causality. The authors identify interesting differences in S25A/D mutant 53BP1 vs WT 53BP1 promoter binding patterns, but how these alters gene expression patterns is hard to discern from the way the data is presented. I also found the evidence backing for the proposed model (and sequence of events) to be lacking. If ATM indeed regulates 53BP1 promoter binding via pS25 phosphorylation, this could be easily demonstrated by ChIP'ing wild type and mutant p53 in the ATM-KO cells, as one would expect the effect to increase promoter binding in the case of the wild-type but the mutant protein would not be impacted. However, even if this was the case, perhaps an even more important gap in the model is the DNA-damage stimulus for ATM-dependent regulation of 53BP promoter interactions? IT is proposed that DDR signalling via ATM be both positively and negatively regulates 53BP1-promoter interactions, via gH2AX-dependent recruitment, and Ser-25 phosphorylation of 53BP1, respectively. However, this is a counterintuitive concept to grasp, and since it is not supported by the included evidence, I'm left not really understanding the mechanism or how it might regulate neuronal identify or function.

Specific Comments:

- As described above, causality is not established in the case of how wild-type 53BP1 promoter interactions are impacted by ATM status, and to what degree this occurs via Ser-25 phosphorylation, thus this conclusion is not substantiated.

- Earlier work showed this 53BP1-depedentn regulation of neuronal function was not conserved between mouse and man. This manuscript should be transparent in stating this is not a well conserved mechanism.

- If this is all dependent on ATM signalling, what is then stimulating ATM activity, and how its this guiding this activity specifically at promoters?

- The author specuflate this might be controlled by gH2AX-dependent recruitment of 53BP1 (presumably to sites of DNA damage). If so, then this would be controlled by gH2AX-MDC1-RNF8-RNF168-H2AK15ub axis which is essential for 53BP1 chromatin binding. RNF168 KO cells would test or refute this hypothesis and such an experiment would be needed to substantiate such a claim

- Line 93-94: "By D55, a subset of ATM-KO cortical organoids appeared smaller than 

controls, but this difference was not statistically significant (Supplementary Fig 3A-B). " - if it not significant this conclusion should be removed

- Line 33-36 "to localize to chromatin with double-stranded breaks, 53BP1 uses its BRCT domain to bind to gH2AX, the Tudor domain to bind to H4K27 dimethylation, and its UDR segment to bind to ubiquitinated H2AK15 (7, 8, 9, 36 10). " is incorrect; 53BP1 tudor recognise H4K20-methylation not this other state.

Reviewer #2: In the manuscript „Phosphorylation of 53BP1 by ATM controls neurodevelopmental programs in cortical brain organoids" by Lim et al., the authors generated ATM-KO brain organoids and found severe neural developmental defects in these organoids. Next, they generated hESC lines, in which 53BP1-S25 cannot be phosphorylated by ATM anymore, by mutating serine 25 to alanine or aspartic acid using CRISPR/Cas9. By generating brain organoids from these cell lines, the authors found that 53BP1-S25A and 53BP1-S25D brain organoids - while different in comparison to the ATM-KO organoids - exhibit abnormal neural development. Lastly, the authors identified direct target genes of 53BP1 by ChIP-seq. Thus, the authors provide mechanistic insight into the role of 53BP1 phosphorylation by ATM during neural development. As this study is very interesting and provides novel insight into the role of ATM and 53BP1 during neural development, it seems to be suitable for publication in PLOS Biology. However, I have some concerns, which need to be addressed before considering the manuscript for publication. Mainly, the phenotypic analysis of the ATM-KO, 53BP1-S25A and 53BP1-S25D brain organoids in terms of analyzed time points (1 to maximum 2 points depending on the mutation) and features/markers studied is not sufficient to draw the conclusions that the authors have drawn. A more thorough phenotypic analysis is needed to understand which neurodevelopmental processes are disturbed in these brain organoids and which contribution ATM and 53BP1 have in this process. For details, please see below.

1) Almost all analyses of brain organoids were performed at d35 with the exception of a few analyses at d55. On which basis has this time point of analyses been chosen? Brain organoids change during development in terms of morphology and cell type composition, so one or two time points of analyses are normally not sufficient. Could the authors please address the following questions? Was this the only time point when the phenotype was present or strongest? How do early-stage brain organoids look like? When did first abnormalities appear? For completeness, it would be good to add this information to the manuscript.

2) In line with the previous question, in d35 ATM-KO organoids, at least in the pictures shown in Figure 1G, no ventricle-like structures are visible. This raises the question, if these structures degenerated during brain organoid development or if they were never present?

3) The ATM-KO brain organoid exhibit many PAX6-positive cells but these cells are not (well) organized in VZ-like structures (not densely packed, no radial organization). This raises several questions. (i) How did the authors identify the VZ (size)? Would ZO-1 stainings helpful to identify the ventricular surface? (ii) Are the Pax6-positive cells apical radial glia (with an apical contact, here ZO-1 stainings could be helpful again) or did they differentiate to a different progenitor type (without apical contact; basal progenitors?)? (iii) It seems so that there are many more Pax6-positive cells in ATM-KO in comparison to WT brain organoids. Is this true? Could the authors please quantify Pax6-positive cells in WT and ATM-KO brain organoids? This seems to also be the case for Ki67. Could the authors also quantify Ki67? These data would be important to better understand the effects of ATM on neural development.

4) In line 92/93 authors claim to observe reduced neuronal differentiation. How do the authors know this? From the images presented in Figure 1G, it seems so that this is true for some ATM-KO clones but not for all. Could the authors validate this by quantifications. Moreover, the authors present just CTIP2 stainings, which is a marker for deep-layer neurons. Could it be that just deep-layer neurons are affected? Why not using a general neuronal marker like NeuN? Would upper-layer neurons at later stages also be affected?

5) How do the authors know that there are lower levels of proliferation in ATM-KO organoids? For this they would need to quantify the number of KI67 positive cells (see above). Information on mitotic (e.g. PH3 positive) cells would also be helpful, as apical radial glia normally divide at the ventricular surface. Is there reduced mitotic cells because there is no ventricular surface or can these cells now divide freely within in the organoid? ZO-1 stainings in combination with PH3 would be helpful.

6) WT and ATM-KO brain organoids seem to have a very different cell type composition. This might bias the transcriptome analyses: Instead of getting mechanistic insight in what ATM-KO is changing within a given cell population, one would get differences which are caused by different cell type composition. Here, single cell RNAseq or enrichment for certain cell populations (e.g. progenitors vs neurons) would be helpful. Another option would be to perform transcriptomics on NPCs, which the authors could easily produce from WT and ATM-KO cells.

7) Brain organoids and KO lines were derived from one cell line (H9/WA09). Would these phenotypes also be observed, if another cell line would be used in this study? Organoid studies normally using more than one cell line to address biological variation.

8) As ATM phosphorylates 53BP1 and therefore at least partially functions through 53BP1 phosphorylation, I would expect 53BP1-S25A and 53BP1-S25D to have similar phenotypes as ATM-KO. However, the phenotypes are quite different, e.g. number and organization of PAX6-positive cells. How do the authors interpret this? Could the authors please add their considerations to the discussion part?

9) The number of CTIP2-positive cells, from the pictures presented in Figure 3D, looks very different between the various 53BP1-S25A and 53BP1-S25D clones. Could the authors please quantify CTIP2-positive cell numbers and compare between 53BP1-S25A and 53BP1-S25D clones and WT?

10) Why are there fewer progenitor cells in 53BP1-S25A and 53BP1-S25D organoids? Do they proliferate less or have a longer cell cycle or is there increased apoptosis or increased neuronal differentiation? As already mentioned above the characterization of brain organoids is very limited and needs to be expanded to understand the effect of 53BP1-S25A and 53BP1-S25D on neural development, especially as the transcriptomic studies seem not provide a clear mechanism.

11) The transcriptome analysis of 53BP1-S25A and 53BP1-S25D brain organoids does not provide a clear explanation of the observed phenotypes. As described in comment 6) might this be due to the different cell type composition between 53BP1-S25A, 53BP1-S25D and WT organoids? Could the transcriptome analysis not also be performed on cultured NPCs, as this would be a clean population and as this would provide a direct comparison to the ChIP-seq data which was performed on these cells anyway?

Minor comments:

1) The legend of Figure 1A is missing.

2) How did the authors confirm that the generation of NPCs were successful?

3) While the authors have checked, if their CRISPR/Cas9-edited cells were still pluripotent, they have not checked, if the karyotype of these cells is still normal.

4) In Figure 1F and Supplemental Figure S2B, why is there a band visible for ATM in case of some ATM-KO cell lines?

5) In Figure 2E, what do the error bars indicate? 

Reviewer #3: In this study, Lim, Djekidel and colleagues investigate the role of ATM and its target 53BP1 in cortical development. After confirming that 53BP1 is a target of ATM on residue S25, they generate a series of KO and KI ES cell lines for these factors and generate cerebral organoids. In both ATM KO and 53BP1 SA and SD, they observe major developmental defects, including proliferation, fate and overall growth. Because ATM expression and 53BP1 phosphorylation increase between ES cells and NPCs in the absence of increased gamma-H2AX, the authors propose that the observed function of 53BP1 phosphorylation is independent of DNA damage. They next use an extensive array of omics methods to better characterise these defects, including transcriptomics, proteomics, phospho-proteomics and ChIP-seq. These experiments identify a series genes and pathways altered by ATM and 53BP1 mutations and demonstrate that 53BP1 phosphorylation controls its position on chromatin. 

While the study of ATM / 53BP1 during cortical development is certainly an important topic, the manuscript falls short of providing novel conceptual advance. The organoid defects are certainly clear but could arise from any type of defect, including organoid specification defects. The multi-omics analysis is impressive, but we are left with large numbers of altered genes and pathways, a lot of them probably as consequences of organoid defects, and none of this information is used to identify how ATM / 53BP1 may affect early neurogenesis. 

Specific comments

1/ The authors suggest that the role of ATM / P-53BP1 during neurodevelopment could be independent of DNA damage, based on absence of increase of gamma-H2AX in NPCs, as compared to ES cells. However, for this suggestion to hold, DNA damage should be looked at in cerebral organoids where the phenotypic analysis was performed. Related to this, do the authors detect increased apoptosis in their organoid models (as suggested by their omics data). 

2/ The fact that mutated organoids are altered is very clear and convincing, but it is hard to grasp what is causing the defect. The organoids are highly disorganized without clear luminal structures. Is organoid specification correct? In figure 3D for example, PAX6 staining appears severely reduced, which could reflect incorrect identity of the organoids. 

3/ In figures 1 and 3, the quantification of the defects should be properly normalized. The fraction of PH3 or Ki67 cells should be quantified out of PAX6+ progenitors, in order to conclude whether progenitor proliferation is affected. For cell fate, it would be good to quantify the fraction of PAX6+ and Ctip2+ cells out of total cells (DAPI) or to present a PAX6/Ctip2 ratio. 

4/ The authors should provide a ZO-1 staining (or similar) to quantify apical surface area and the formation (or absence of formation) of lumen. 

5/ The PAX6 staining appears very different between ATM KO and 53BP1 KI conditions. Does this reflect different molecular alterations?

---

## [Decision Letter · Decision Letter 2]

4 Jul 2024

Dear Dr Peng,

Thank you for your patience while we considered your revised manuscript "Phosphorylation of 53BP1 by ATM controls neurodevelopmental programs in cortical brain organoids" for publication as a Research Article at PLOS Biology. This revised version of your manuscript has been evaluated by the PLOS Biology editors, the Academic Editor and the original reviewers.

Based on the reviews and on our Academic Editor's assessment of your revision, we are likely to accept this manuscript for publication, provided you satisfactorily address the remaining points raised by the reviewers. In particular, we share Reviewer 1's concerns about the strength of the data implicating HH. We think that it is beyond the scope of the manuscript to work out exactly how ATM is activated in this context, but that the interpretation be toned down, given that the conclusion is based on only a slight difference in the variance of the Western blot data that influences the statistics. Therefore, we suggest that you modify the text to reflect this uncertainty.

Please also make sure to address the following data and other policy-related requests.

* We would like to suggest a different title to improve accessibility: "Phosphorylation of the DNA damage repair factor 53BP1 by ATM kinase controls neurodevelopmental programs in cortical brain organoids"

* Please add the links to the funding agencies in the Financial Disclosure statement in the manuscript details

* DATA POLICY:

1DHI, 2ABCE, 3E, 4CF, 5ABCEFGHI, S1I, S2E and similar panels in the supplementary figures

We require the original, uncropped and minimally adjusted images supporting all blot and gel results reported in an article's figures or Supporting Information files. We will require these files before a manuscript can be accepted so please prepare and upload them now. Please carefully read our guidelines for how to prepare and upload this data: https://journals.plos.org/plosbiology/s/figures#loc-blot-and-gel-reporting-requirements

We expect to receive your revised manuscript within two weeks. 

*Published Peer Review History*

*Press*

Sincerely,

Christian

Christian Schnell, PhD

Senior Editor

cschnell@plos.org

PLOS Biology

Reviewer remarks:

Reviewer #1: The manuscript remains descriptive, since the authors are still unable to explain how the ATM-gH2AX-RNF168-53BP1 is activated, and how/why this non-canonical role of DNA damage signaling is controlling specific genetic programs in NPCs and cortical organoids, while in other somatic tissues the gH2AX-RNF169-53BP1 pathway is a well-established to specifically regulate DNA repair. They have tried to address the activating stimulus of ATM, and in new experiments with cyclopamine (HH inhibitor) they find reduced pS25-53BP1 levels and suggest that in cortical neurons, hedgehog signalling is likely required to activate ATM. Again, HH would be an unexpected and non-canonical activator of ATM, and the claimed alterations of pS25-53BP1 levels in new immunoblot data in new Supp Fig 16 aren't clear-cut enough to convince me that these conclusions have sufficiently substantiated evidence basis. The author rebuttal states that "the exact clarification of mechanisms promoting ATM activities, especially in directing its kinase activity at specific promoters, is beyond the scope of this study. " yet nevertheless they also state that this study "has broad implications about diseases associated with ATM, including ataxia telangiectasia". Although, I acknowledge that the authors have attempted to address my concerns with new experiments to explore the role of gH2AX, and add new correlative data regarding RNF168, there absence of solid evidence to explain the mechanism basis of this non-canonical role of the ATM-gH2AX-53BP1 axis, and so I remain cautiously sceptical of the proposed mechanisms and their overall impact.

Reviewer #2: The authors addressed all my comments and improved the manuscript substantially. I can recommend the current version of the manuscript for publication in PLOS Biology.

Reviewer #3: The authors have now addressed all of my previous concerns: the absence of DNA damage increase with differentiation, the correct specification of mutant organoids (comments 1 and 5), the normalization method for the quantifications, and the use of ZO-1 staining to quantify the lumen defects.

---

## [Editor Report · Decision Letter 3]

19 Jul 2024

Dear Dr Peng,

Thank you for the submission of your revised Research Article "Phosphorylation of the DNA damage repair factor 53BP1 by ATM kinase controls neurodevelopmental programs in cortical brain organoids" for publication in PLOS Biology. On behalf of my colleagues and the Academic Editor, Madeline Lancaster, I am pleased to say that we can in principle accept your manuscript for publication, provided you address any remaining formatting and reporting issues. These will be detailed in an email you should receive within 2-3 business days from our colleagues in the journal operations team; no action is required from you until then. Please note that we will not be able to formally accept your manuscript and schedule it for publication until you have completed any requested changes.

While you are attending to the formatting and production requests to come, we would also like you to state the limitations of the data shown in Fig S16 more clearly. We are not convinced that it is accurate to single out HH signaling, and it would be rather more accurate to simply say that the data suggest potential effects of those signaling pathways but more experiments would be needed. 

PRESS

Sincerely, 

Christian

Christian Schnell, PhD 

Senior Editor

PLOS Biology

cschnell@plos.org